# Measuring the olfactory bulb input-output transformation reveals a contribution to the perception of odorant concentration invariance

Douglas A. Storace[1] & Lawrence B. Cohen[1,2]

Humans and other animals can recognize an odorant as the same over a range of odorant concentrations. It remains unclear whether the olfactory bulb, the brain structure that mediates the first stage of olfactory information processing, participates in generating this perceptual concentration invariance. Olfactory bulb glomeruli are regions of neuropil that contain input and output processes: olfactory receptor neuron nerve terminals (input) and mitral/tufted cell apical dendrites (output). Differences between the input and output of a brain region define the function(s) carried out by that region. Here we compare the activity signals from the input and output across a range of odorant concentrations. The output maps maintain a relatively stable representation of odor identity over the tested concentration range, even though the input maps and signals change markedly. These results provide direct evidence that the mammalian olfactory bulb likely participates in generating the perception of concentration invariance of odor quality.

[1] Department of Cellular and Molecular Physiology, Yale University School of Medicine, 333 Cedar Street, New Haven, Connecticut 06520, USA. [2] Center for Functional Connectomics, Korea Institute of Science and Technology, Seoul 136-791, Republic of Korea. Correspondence and requests for materials should be addressed to D.A.S. (email: dstorace@gmail.com) or to L.B.C. (email: lawrence.b.cohen@hotmail.com)

Understanding how sensory objects can be identified over a range of stimulus intensities remains a fundamental question in neuroscience. In the olfactory bulb, thousands of olfactory receptor neurons each expressing the same receptor protein converge onto one or two regions of bulb neuropil called glomeruli. There these cells synapse onto the apical dendrites of a few dozen mitral and tufted cells, which only innervate that glomerulus, and whose axons provide all of the output to higher brain regions. Thus, the olfactory bulb's input and output are defined anatomically and they spatially overlap in glomeruli[1, 2].

While it is often considered that odor identity is determined by the combination of odorant receptors that are activated by an odorant[3], in its simplest form this hypothesis is contradicted by the spatial patterns of input activity across glomeruli (i.e., activity maps). The maps of the input to the olfactory bulb and maps of the glomerular intrinsic signals are a confound of odorant identity and concentration[4–9]. The maps changed when the odorant was changed, but the maps also changed qualitatively when odorant concentration was changed. Despite the olfactory bulb receiving this seemingly ambiguous signal, humans and other animals can recognize an odorant as the same over a range of odorant concentrations[10–14]. It was unclear where this perceptual invariance is generated in the olfactory pathway. The mitral and tufted output cells directly innervate 12 different brain regions in the mouse[15], and these in turn activate other brain areas. Some of these higher brain regions, such as the piriform cortex, are thought to perform computations in which intensity-invariant responses are important[16].

Several studies have speculated that the olfactory bulb may participate in generating the perception of concentration invariance[17, 18]. Individual output cells are sensitive to concentration changes and thus are unlikely candidates to encode odor identity. However, their distributed activity across a population encodes information about odor identity across a range of concentrations[19–21]. A number of studies have demonstrated bulbar mechanisms that are candidates for generating stable intensity-invariant odor responses[22–25]. However, a direct comparison of input and output maps has not been reported.

Here we compared the glomerular odor activation patterns of the bulb output with its input. Some input/output measurements were carried out on the same glomeruli in one hemi-bulb, while others were carried out in opposite hemi-bulbs in the same animal. In yet other input/output transformation measurements, the inputs and outputs were measured in separate preparations. In contrast with the input, the output spatial activation patterns were relatively similar to each other across odor concentrations, although the glomerular output amplitude still encoded odor concentration. These results show that the olfactory bulb removes some of the confound of odorant concentration on the input activity maps so that odorant identity is represented by the output maps, while still maintaining sufficient concentration dependence to encode intensity differences.

## Results

**Approach for imaging input and output**. We developed an approach for determining the input/output transformation of the mammalian olfactory bulb. It involves measuring both the input and output of individual glomeruli. Two different sensors of neural activity were used: one in the olfactory receptor neurons (input) and the other in the mitral and tufted cells (output). This was accomplished by employing a combination of anatomical and genetic targeting. Olfactory receptor neurons were anatomically targeted via nasal infusion with an organic calcium sensitive dye (Fig. 1a)[4, 7, 8]. In the same preparation, genetically encoded voltage or calcium indicators (GEVIs or GECIs) were targeted to

mitral and tufted cells using either adeno-associated virus (AAV) transduction with cre-dependent vectors in a transgenic mouse that expresses cre recombinase in mitral and tufted cells (*Protocadherin 21*-Cre) (Fig. 1a)[26], or a transgenic mouse that expresses GCaMP6f selectively in mitral and tufted cells (*Thy1*-GCaMP6f)[27]. Histological examination confirmed that the sensors were in the expected locations (Fig. 1b; Supplementary Fig. 1).

By using input and output sensors that have substantially different spectral properties, input and output signals could be separately measured in the same glomeruli in the same hemi-bulb by changing the excitation and/or emission wavelength(s). Wide-field epi-fluorescence imaging was used to measure odor-evoked activity across ~2 log units of odorant concentration in freely breathing anesthetized mice. Input measurements were made using either the calcium dye Fura dextran or Cal-590 dextran[28, 29]. Output measurements were made using the protein voltage or calcium indicators ArcLight, GCaMP6f, or jRGECO1a[30–34]. The Fura dextran and Cal-590 dextran input signals were imaged using 380 and 572 nm excitation light, respectively. The ArcLight and GCaMP6f output signals were imaged with an excitation light of 479 nm. The jRGECO1a output signals were imaged using an excitation light of 565 nm. Fluorescence time courses were recorded from regions of interest (ROIs) corresponding to the activated glomeruli. These time courses reflect the population average of the olfactory receptor neuron axon input activity or the average of the mitral and tufted neuron output activity from individual glomeruli[4, 8, 35, 36].

**Same glomeruli comparisons**. Measurements from a single glomerulus using Fura dextran to monitor calcium input signals and ArcLight to monitor membrane potential output signals showed that the amplitude of the odor-evoked input signal was a relatively steep function of odorant concentration (Fig. 1c, input), a result that is consistent with the prior reports[4, 5, 7]. In contrast, the output signal amplitude varied less as a function of concentration (Fig. 1c, output). This result from one glomerulus suggests that the output is more concentration invariant than the input, but does not show whether the output map remains a confound of odorant concentration and identity.

In a second preparation, signal measurements were made from 13 glomeruli at four odorant concentrations using Fura dextran for the input signals and GCaMP6f for the output signals. Activity maps for the input and output were made by subtracting imaging frames prior to the stimulus from frames acquired during the odorant response (Fig. 2a, timing of the selected frames is indicated by the black bars below the input signal in the sixth pair in Fig. 2c). The output maps at the four odorant concentrations are more similar to each other than are the input maps. Input and output map similarity across odor concentrations was quantified by measuring the spatial correlation of the map evoked by each odor concentration with the maps evoked by each of the other odor concentrations (Fig. 2b). The time course of the input and output signals from the 13 glomeruli (Fig. 2a, ROIs) again demonstrate that the input signal is a considerably steeper function of odor concentration than the output signal (Fig. 2c). For several glomeruli, there was no detectable input signal at the lowest odorant concentration (i.e., glomeruli 2, 5, and 6) while the output signal remained. The amplitudes of the odor-evoked input and output signals for each glomerulus were normalized to the response elicited by 11% of saturated vapor, and are plotted as the individual thin lines in Fig. 2d. The mean evoked signals are shown in the thick lines (red for output; black for input). A decrease in odor concentration from 11 to 0.12% caused the

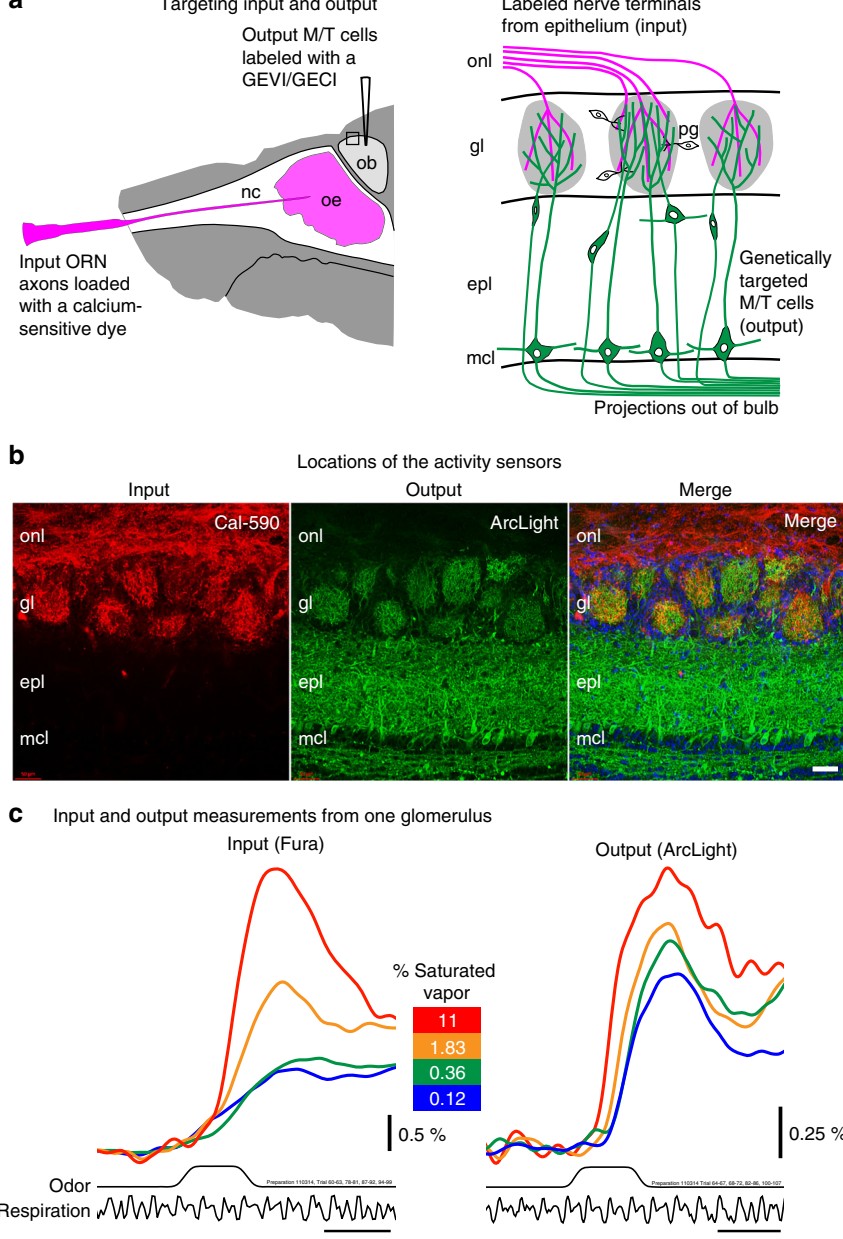

**Fig. 1** Cell specific targeting of activity sensors to the olfactory receptor neuron input and the mitral/tufted cell output, and optical measurements from one glomerulus. **a** Experimental approach: (*left*) Olfactory receptor neuron nerve terminal input was labeled via a nasal infusion of a calcium sensitive dextran dye. Cre recombinase expressing mitral and tufted cell output was targeted using cre-dependent viral vectors that expressed genetically encoded voltage or calcium indicators (GEVI or GECI). (*right*) By using activity sensors with substantially different excitation or emission spectra, input vs. output can be measured independently from the same glomerulus by changing the excitation or emission wavelengths. **b** A histological section shows anatomical targeting of Cal-590 dextran to the olfactory receptor nerve terminal input in the glomeruli (*left*) and genetic targeting of ArcLight to the mitral and tufted cell output (*middle*) in the same section along with a merged image containing DAPI (*right*). **c** Optical measurements from input and output sensors in one glomerulus in response to ethyl tiglate presented across ~2 log units of odorant concentration (0.12–11% of saturated vapor). Input and output measurements were performed sequentially using Fura dextran and ArcLight, using excitation wavelengths of 380 nm and 480 nm, respectively. The optical traces are low-pass filtered at 1 Hz and are the average of 3–7 individual trials aligned to the first sniff following odor onset. The odor and respiration traces are from one of the single trials. The data in **b** and **c** are from different preparations. onl, olfactory nerve layer; gl, glomerular layer; epl, external plexiform layer; mcl, mitral cell layer, ORN, olfactory receptor neuron; M/T, mitral and tufted cells. Scale bar in b, 50 µm. Details for the data in panel **c** is included in Supplementary Table 2

input to decrease by $88 \pm 1\%$, whereas the output decreased by only $51 \pm 4\%$ (Fig. 2d, $p < 0.001$). Similar $p$ values were observed for the comparisons at 0.36 and 1.83% of saturated vapor (0.36%: $p < 0.001$; 1.83%: $p < 0.001$; detailed statistics are included in Supplementary Table 2).

The result that the output maps and glomerular time course amplitude changed much less than the input with changes in concentration was confirmed in three additional preparations using either Fura and ArcLight (Supplementary Fig. 2; 0.36%: $p = < 0.001$; 0.12%: $p < 0.001$), Cal-590 and

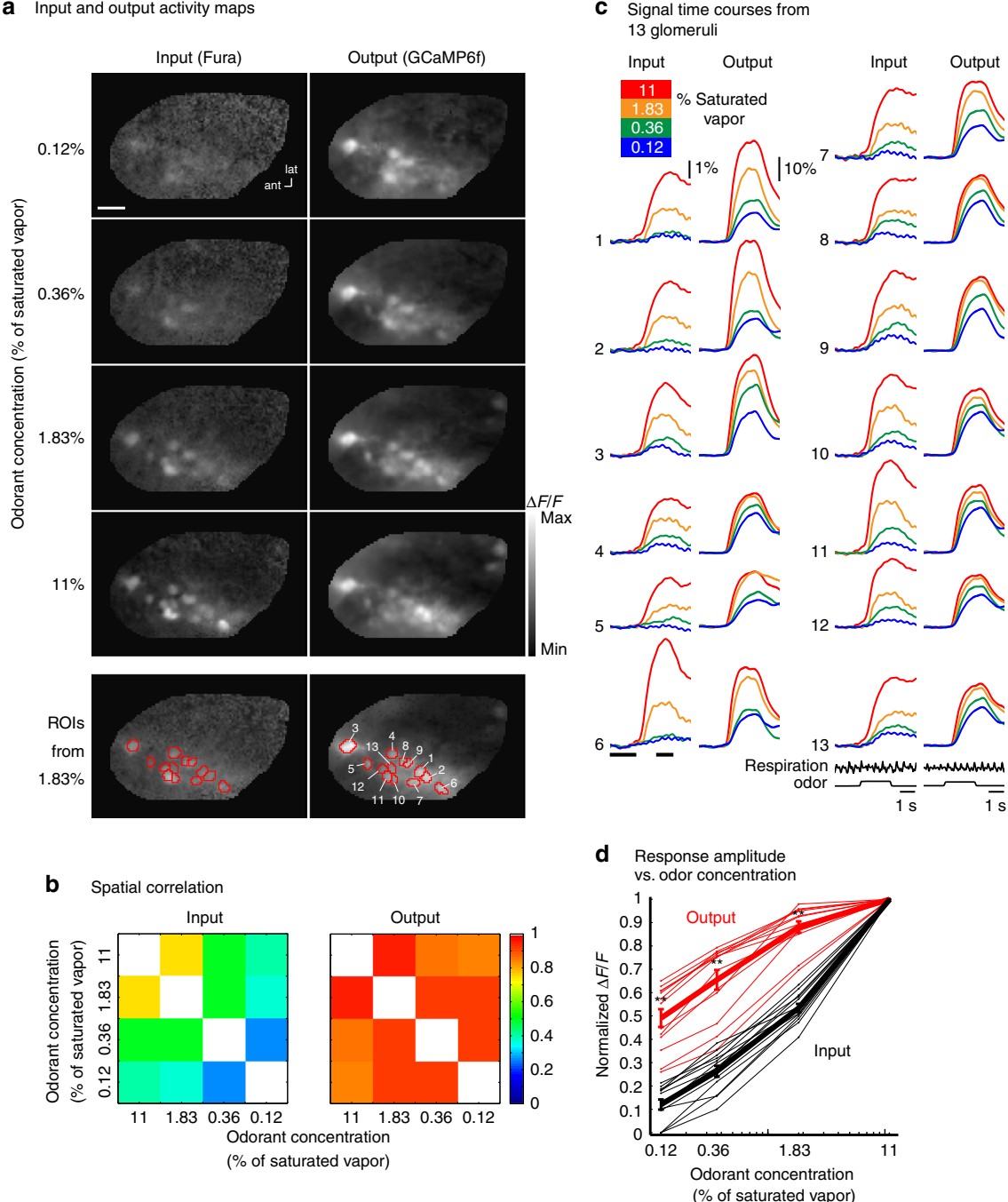

**Fig. 2** Comparing the input and output maps and input and output signals from 13 glomeruli in the same bulb. **a** Frame subtraction maps of input (*left*) and output (*right*) activity at four odorant (methyl valerate) concentrations. The output maps are substantially more similar to each other than the input maps are to each other. The bottom panel (ROIs from 1.83%) indicates the selected glomeruli (*red overlay*) for the time course analysis in **c**. The largest glomerular signals (Δ*F/F*) for each map are from low to high concentration: (*input*) 0.9%, 1.6%, 3%, 6%; (*output*) 17.3%, 28.4%, 34.8%, 45.2%. **b** Spatial correlation of each odor concentration frame subtraction map with the maps from each of the other odor concentrations. The output map correlations are much more similar to each other than are the input maps. **c** Traces of fluorescence vs. time for each of the 13 glomeruli (input on *left*, output on the *right*). The input signals decrease more dramatically than those of the output. Input and output measurements were performed sequentially using Fura dextran and GCaMP6f, using excitation wavelengths of 380 nm and 480 nm respectively. The black bars underneath the input trace for glomerulus 6 indicate the time points used to generate the activity maps in **a**. The traces are low-pass filtered at 1 Hz. The odor and respiration traces are from single trials. The activity maps and traces are from the same data set and are from an average of 4–6 individual trials that were aligned to the first sniff following the odor onset. **d** *Thin lines*: normalized peak fluorescence change vs. odorant concentration for the 13 glomeruli in **c**. *Thick lines*: mean of the 13 glomeruli. Signal size was normalized to the response elicited by 11% of saturated vapor for each glomerulus. The error bars represent s.e.m., **$p < 0.005$ (Wilcoxon rank sum). Scale bar in **a**, 250 μm. An air alone trial was subtracted from the input for the traces and values in **c** and **d**. Detailed statistical results are in Supplementary Table 2

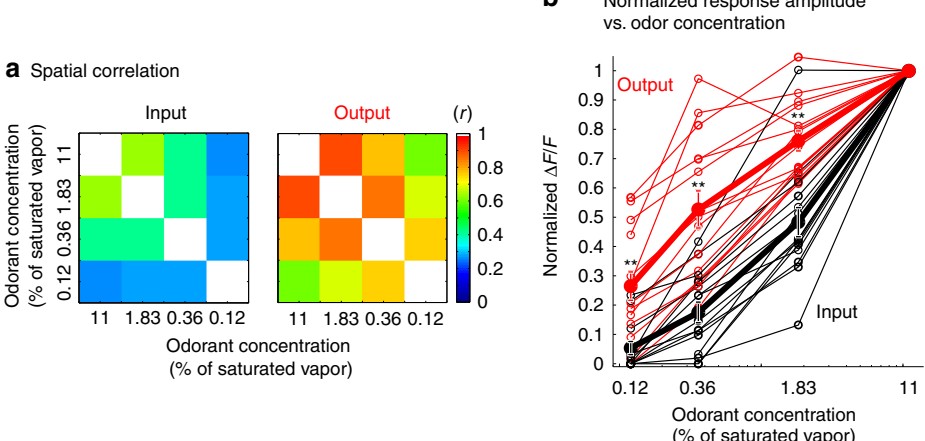

**Fig. 3** Population summary of the spatial correlation of the activity maps and signal size vs. odor concentration. **a** The mean spatial correlation of each frame subtraction map with the map from the other three concentrations. The activity maps of the output (*right*) are more similar to each other than are the activity maps of the input (*left*). This summary includes 14 measurements from 13 preparations (responses to two different odorants were measured in one preparation). **b** Normalized signal size vs. odorant concentration for 15 measurements in 14 preparations (*thin lines*, two different odor responses were measured in one of the preparations). The average of the 15 measurements is shown as the *thick lines*. The input signals (*black*) decline much more rapidly as the odor concentration is reduced. In same hemi-bulb preparations, all activated glomeruli that could be identified for both input and output were included. In opposite bulb preparations, input and output glomeruli were counted if they were activated by 1.83% of saturated vapor. Panel **a** includes 13 measurements using methyl valerate and 1 using isoamyl acetate. Panel **b** includes 1 measurement using ethyl tiglate, 13 using methyl valerate, and 1 using isoamyl acetate. The results from the different odorants were not qualitatively different. The error bars represent s.e.m., **$p < 0.005$. A repeated measures ANOVA and Wilcoxon rank sum test were used for the statistical analysis. Detailed experimental and statistical information for panels **a** and **b** are included in Supplementary Tables 1 and 2, respectively

GCaMP6f (Supplementary Fig. 3; 1.83%: $p < 0.001$; 0.36%: $p < 0.001$; 0.12%: $p < 0.001$) or Fura and jRGECO1a (Supplementary Fig. 4, 1.83%: $p < 0.01$; 0.36%: $p < 0.001$; 0.12%: $p < 0.001$) (detailed statistics are included in Supplementary Table 2).

**Summary of same glomeruli and same animal comparisons.** We made measurements in 14 preparations (including the five used in Figs 1 and 2 and Supplementary Figs 2–4) using different combinations of input (Fura dextran, Cal-590 dextran, Oregon Green 488 BAPTA-1 dextran, and Calcium Green-1 dextran) and output (ArcLight, GCaMP6f, GCaMP6s, and jRGECO1a) sensors that had varying calcium affinities and Hill coefficients. Measurements with sensor combinations that did not meet the spectral requirement were made in opposite hemi-bulbs in the same preparation. Because glomerular responses are similar (but not identical) across hemi-bulbs[37], this type of measurement provides information about the population behavior of input vs. output glomeruli. The comparison is less direct since the input and output of the same glomeruli are not compared.

The spatial correlations of the output activity maps are significantly more correlated with each other than are the input maps (Fig. 3a; $p < 0.05$ for all comparisons, detailed statistics for each comparison is included in Supplementary Table 1a). The normalized response amplitude of input and output glomeruli that were activated by odorant presentation was analyzed in the 14 preparations. A total of 142 input glomeruli ($9.5 \pm 1.2$ per preparation) and 214 output glomeruli ($14.1 \pm 2.3$ per preparation) were analyzed. The normalized output response was significantly larger than the normalized input in most of the individual preparations (Fig. 3b, thin red and black lines; the individual values are included in Supplementary Table 2). The normalized input and output response for each odor concentration was averaged across the identified glomeruli for each preparation. There was a significant effect of input–output

($F(1, 11) = 27$, $p < 0.001$) and odor concentration ($F(2, 22) = 171$, $p < 0.001$). Input-normalized and output-normalized signal sizes were compared at each odor concentration. The output was significantly larger than the input at all tested concentrations after adjusting the significance threshold for multiple comparisons (Fig. 3b; thick red and black lines; $p < 0.005$; detailed statistics are included in Supplementary Table 2).

The slope of the mean concentration-response function for each preparation (Fig. 3b, *thin lines*) was estimated by fitting the data to a modified form of the Hill equation[4, 38]. The mean Hill coefficients for the input and output were $1.57 \pm 0.13$ (range: 0.8–2.6, $N = 15$) and $0.96 \pm 0.1$ (range: 0.5–1.5, $N = 15$), respectively. These values corresponded well to prior reports of glomerular input and output[4, 5, 39].

There are diffuse optical signals in the input[4] and output measurements. The diffuse output signals likely originate from the out of focus mitral and tufted cell lateral dendrites when using epifluorescence imaging[35, 36]. We carried out three different analyses in an attempt to determine whether the differences between input and output were affected by the diffuse signal. First, the correlation analysis in Fig. 3a was repeated using only the pixels containing the glomerular ROIs used for the signal size measurements in Fig. 3b. This yielded a similar input/output difference and a similar statistical significance for all comparisons (Supplementary Table 1b; $p < 0.05$). Second, although spatially high-pass filtering the output maps slightly reduced the mean spatial correlation for each comparison (between $r = 0.03$ and $r = 0.13$), the results were not significantly different from the unfiltered maps ($p > 0.05$ for all comparisons). Third, we estimated the diffuse signal for both input and output as the $\Delta F/F$ from the parts of the dorsal surface of the bulb that did not contain glomerular sized peaks of activity and subtracted the diffuse signal from the signal measured from the glomerular ROIs at each odor concentration (Supplementary Fig. 5a, b). The corrected values for each glomerulus were normalized to the corrected value evoked by 11% of saturated vapor for each

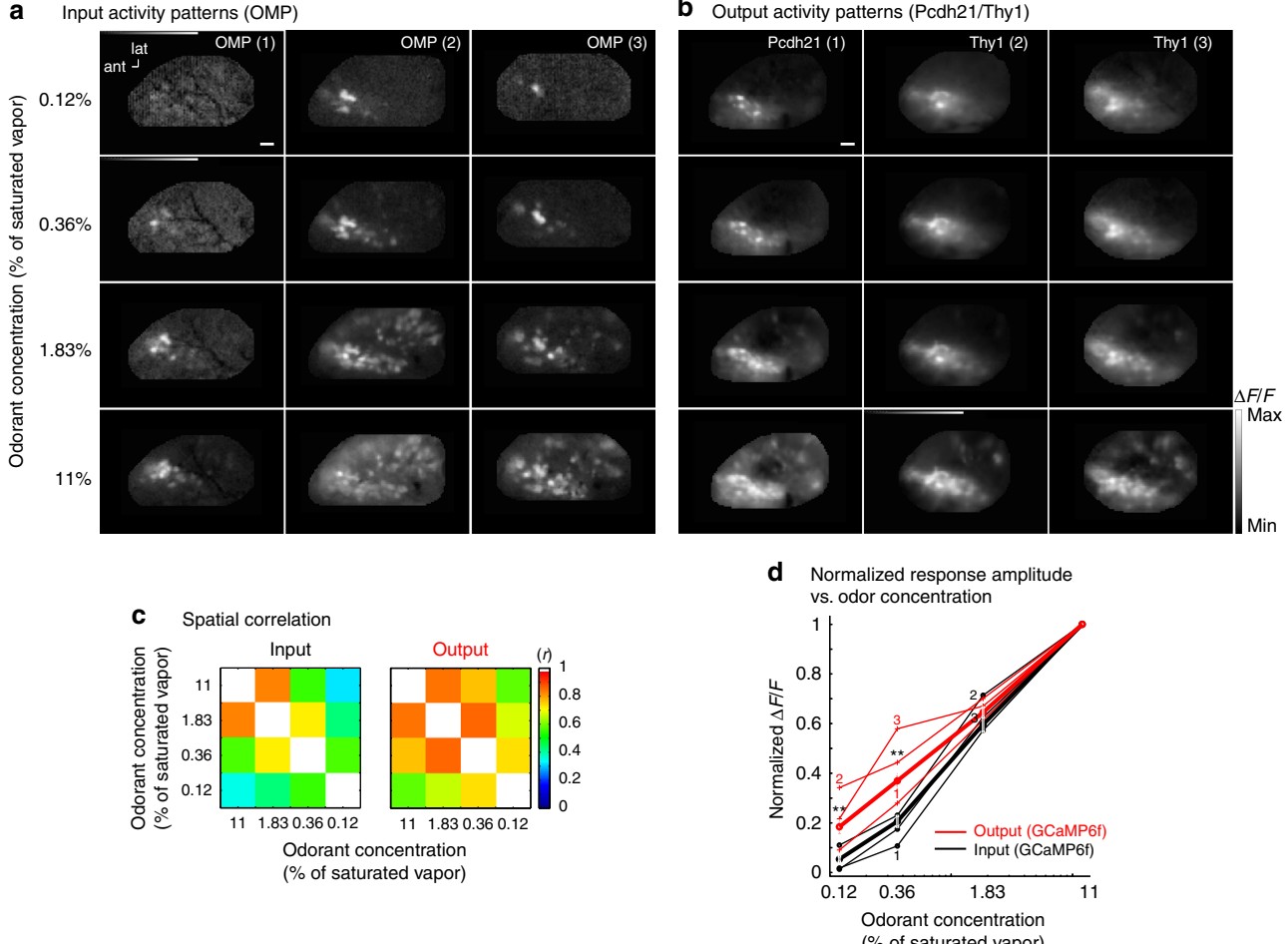

**Fig. 4** Comparison of GCaMP6f in input and output in separate preparations. **a** Frame subtraction maps of activity from three different preparations expressing GCaMP6f in the input olfactory receptor neurons (OMP-GCaMP6f mice). The largest glomerular signals (ΔF/F) for each preparation are from low to high concentration (*left*) 0.6%, 2.6%, 5.6%, 10%; (*middle*) 4%, 5.9%, 9%, 11%; (*right*) 1.2%, 4.4%, 6.5%, 7.2%. **b** Frame subtraction maps of activity from three different preparations expressing GCaMP6f in the output (Pcdh21-Cre or Thy1-GCaMP6f mice). The largest glomerular signals (ΔF/F) for each map are from low to high concentration: (*left*) 10%, 27%, 46%, 59%; (*middle*) 27%, 29%, 45%, 52%; (*right*) 15%, 38%, 44%, 51%. **c** Mean spatial correlation of each odor concentration frame subtraction map with the maps from each of the other odor concentrations. Correlation differences were assessed using a Mann-Whitney U-Test: 11 vs. 1.83% ($U = 66$, $p = 0.07$), 11 vs. 0.36% ($U = 34$, $p = 0.001$), 11 vs. 0.12% ($U = 58$, $p = 0.03$), 1.83 vs. 0.36% ($U = 40$, $p = 0.003$), 1.83 vs. 0.12% ($U = 50$, $p = 0.01$), 0.36 vs. 0.12% ($U = 55$, $p = 0.02$). **d** Normalized amplitude vs. odor concentration. The *thin lines* are from the example preparations in panels **a** and **b**. The *thick lines* are averages that include input measurements from 11 hemibulbs in 7 preparations, and output measurements from 22 hemibulbs in 18 preparations (Pcdh21-Cre: 14 hemibulbs from 14 preparations including the 7 preparations in Fig. 3b; Thy1-GCaMP6f: 8 hemibulbs from 4 preparations, including the preparation in Supplementary Fig. 3). The error bars represent s.e.m., *$p < 0.05$; **$p < 0.01$ (Kruskall-Wallis Test, Mann-Whitney U-Test). Scale bars in **a**, **b**, 250 μm. ant, anterior; lat, lateral

preparation (Supplementary Fig. 5c). The corrected and uncorrected normalized response amplitude across all glomeruli for each preparation was not significantly different from each other at any concentration (Supplementary Fig. 5d; $p > 0.7$). These results suggest that the diffuse signal does not substantially alter our result that the output signals change less than the input with odor concentration changes.

**Same sensor but different preparations.** In order to compare input and output measurements made with the same sensor, the input and output measurements had to be made in separate preparations. We carried out the input measurements in transgenic mice that express GCaMP6f in the olfactory receptor cell input (OMP-GCaMP6f) (Fig. 4a; Supplementary Fig. 1a–c; 11 hemi-bulbs in 7 preparations). The GCaMP6f output data include the 8 GCaMP6f preparations from Fig. 3b (8 hemi-bulbs),

7 output-only preparations from Pcdh21-Cre transgenic mice (7 hemi-bulbs), and 3 additional Thy1-GCaMP6f transgenic mouse preparations (6 hemi-bulbs). Activity maps from three preparations are shown for both input and output (Fig. 4a and b). The output maps were significantly more correlated with each other than the input for most comparisons (Fig. 4c, $p < 0.05$ for all comparisons except the correlation between 11 and 1.83% of saturated vapor). There was a significant effect of input–output ($X^2(1) = 5$, $p < 0.05$) and concentration ($X^2(2) = 72$, $p < 0.001$) on normalized signal size. There was a significant difference between input and output at 0.36% ($U = 29$, $p < 0.001$) and 0.12% ($U = 33$, $p < 0.001$) of saturated vapor. Thus, in this comparison where the same sensor was used, both the input and output activity patterns, map correlations, and signal amplitudes as a function of concentration were similar to the results presented in Fig. 3 where different sensors were used but the comparison was made in individual mice. The slopes of the mean concentration-response

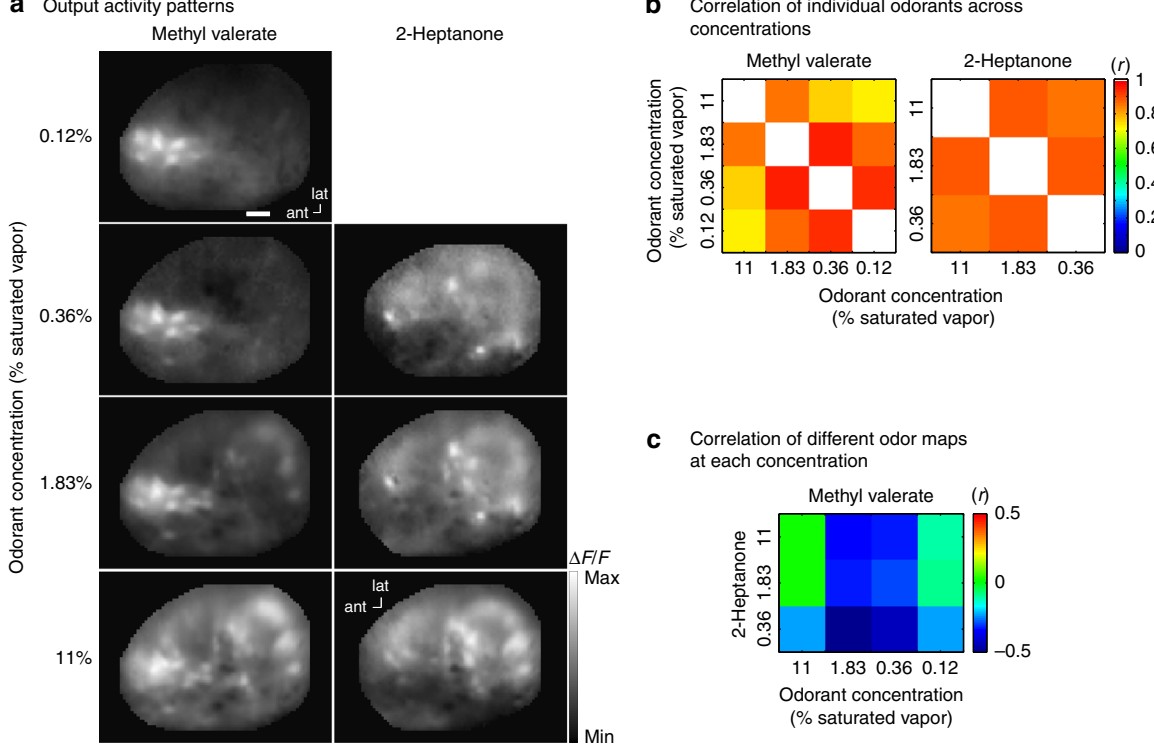

**Fig. 5** Output maps are relatively concentration invariant, but odorant specific. **a** Maps of the output in response to the two odorants methyl valerate and 2-heptanone presented at a range of concentrations. The largest signals ($\Delta F/F$) from lowest to highest concentration for methyl valerate: 17%, 22%, 27%, 46%; 2-heptanone 13%, 22%, 50%. **b** Correlation of output maps evoked by the same odorant across a range of concentrations. **c** Correlation of the output maps evoked by two different odorants across a range of concentrations. This example is from a Thy1-GCaMP6f transgenic mouse in which GCaMP6f is selectively expressed in bulb output neurons[27] (Supplementary Fig. 1g–i). Similar results were obtained in two other preparations using Pcdh21-Cre transgenic mice with AAV1 GCaMP6f injections when comparing methyl valerate to isoamyl acetate, or methyl valerate to 2-hexanone. Scale bar in **a** 250 µm. ant, anterior; lat, lateral

function for input and output measured in different preparations were similar to those measured in the same preparations. The mean Hill coefficients for the input and output were $1.3 \pm 0.06$ (range: 1.1–1.7, $N = 11$) and $0.94 \pm 0.05$ (range: 0.5–1.4, $N = 22$), respectively.

**Comparing the input measured with an organic dye and a GECI.** We also compared the input activity maps measured using either GCaMP6f or an organic calcium dye in the same hemi-bulb. The red-shifted calcium dye Cal-590 dextran was nasally loaded into one hemi-bulb of OMP-GCaMP6f transgenic mice (two hemi-bulbs in two preparations). The input activation patterns measured using the organic dye and protein sensor were similar (Supplementary Fig. 6). This result indicates that both organic calcium dyes and protein calcium sensors report maps of glomerular input that are similarly concentration-dependent.

**Odor identity.** Is odor identity conserved across concentration in the output maps? Output measurements were performed in response to different odorants at different concentrations in three preparations. For a particular odorant, the spatial activation patterns were visually similar across concentrations (Fig. 5a) and were highly correlated (Fig. 5b) ($r$ ranging from 0.35 to 0.9), consistent with the results in Figs 2 and 3. However, different odorants evoked distinct activation patterns (Fig. 5a, compare methyl valerate and 2-heptanone) that were poorly correlated (Fig. 5c). The correlations of the maps evoked by different odorants ($r$ ranging from 0.03 to 0.35) were always less than the

correlation evoked by the same odorant at different concentrations. Thus, odorant identity appears to be conserved across concentration changes in the bulb output.

The results obtained when using the same protein sensor in the input and output, albeit in different preparations (Fig. 4) were similar to those in experiments with different sensors in input and output in the same preparation (Fig. 3). In addition, using both an organic dye and a protein sensor for measuring input yielded similar input maps (Supplementary Fig. 6). These results show that the differences between input and output seen with different sensors cannot be explained by differences in indicator properties.

**Discussion**

It has been proposed that odor identity is determined by the combination of odorant receptors that are activated by an odorant. However, the glomerular maps of the input to the olfactory bulb are a confound of odorant identity and odorant concentration[4, 5, 7–9], a result we have confirmed (Fig. 2, Supplementary Figs 2–4). Our results show that odorant identity is more likely determined by the glomerular output of the olfactory bulb (Fig. 5). The olfactory bulb in large part removes the qualitative effect of odorant concentration so that the output maps mainly represent odorant identity, although the population average activity signal of the mitral and tufted cells connected to a glomerulus still encodes intensity differences. However, although the output maps across concentrations are more similar to each other than are the input maps, they are not identical. Other brain

regions could remove these differences, or pattern completion functions elsewhere might be sufficient so that this level of identity is good enough.

The result that the output maintains a relatively stable odor representation over a range of concentrations is consistent with populations of individual mitral cell recordings that remain correlated over a range of concentrations[19–21]. A stable representation of odor identity over a range of concentrations could be useful for upstream processing since odor identity would be defined by the specific pattern of mitral cell output. This function is likely important for generating the perception that the quality of an odorant is considered the same over a range of concentrations. That said, proving that the olfactory bulb contributes to the perception of concentration invariance would require a manipulation that is specific for disrupting only the mechanism(s) responsible for the concentration-invariant bulb output.

Determining the mechanism(s) behind the transformation carried out by the olfactory bulb is likely to be difficult because the olfactory bulb input-output transformation is shaped by presynaptic mechanisms[40–43], more than 20 different interneuron cell types in the glomerular layer[44], granule cell interneuron processing, as well as feedback from several brain structures[45]. Dopaminergic/GABAergic, parvalbumin and granule cell interneurons are all likely contributors as they form local[46, 47] and long-range connections[25, 48, 49] that can modulate olfactory bulb output. Cleland et al.[17, 18] have proposed that these types of interneurons could act to implement either a gain control or normalization of the output via synapses onto external tufted cells[17, 18, 50]. This idea is supported by the recent demonstration of subpopulations of interneurons that can boost responses to weak stimuli via electrical synapses, while attenuating responses to strong stimuli via GABAergic synapses[24, 25, 51].

The type of preparation used for measuring input and output signals depends on the sensors that are used. When the input and output sensors had minimal spectral overlap, we could measure both the input and output from the same glomeruli. When the sensor spectra overlapped we performed the experiment in opposite hemi-bulbs in the same mice (Supplementary Table 2). When we used transgenic mice expressing the same sensor in either the olfactory receptor neurons or the mitral/tufted cells, the input and output measurements were carried out in separate preparations (Fig. 4).

The out-of-focus lateral dendrites of the mitral and tufted cells can contribute a diffuse signal that lacks glomerular specificity when imaging using wide-field epifluorescence[35, 36, 39]. A diffuse signal was also known to be present in the input[4]. Although a non-selective signal could alter our input/output measurements, we think that this is unlikely to be the cause of the difference between input and output. A diffuse signal emanating from lateral dendrites would not result in glomerular sized peaks of activity. In addition, our analyses show that the differences between the input and output correlations using only the glomerular ROIs were similar (Supplementary Table 1a–b), high-pass spatial filtering did not significantly reduce the correlations from the output ROIs, and that subtracting the non-glomerular diffuse signal does not substantially alter the input or output's concentration dependence (Supplementary Fig. 5).

Sensors have different biophysical properties that include calcium affinity ($K_d$), binding kinetics (Hill coefficient) and binding rates. We used organic calcium dyes and protein sensors with similar properties (Supplementary Table 3), although the protein sensors tended to have higher Hill coefficients[30, 32–34, 52–58]. In principle this should yield a steeper calcium-dependent relationship, which makes it possible that the output may vary even less with concentration than indicated by our results[59]. Furthermore, the similar results obtained using the same protein sensor in both

input and output in separate preparations (Fig. 4), and the input measurements from the same bulb using both an organic calcium dye and protein sensor (Supplementary Fig. 6) strongly suggests that our results cannot be explained by sensor differences.

The limited signal-to-noise ratio in the organic dye measurements results in a noise level that is ~2–5% of the largest signal. We speculate that if the largest signal represents the activation of ~1000 olfactory receptor neurons, then the smallest detectable signal would be equivalent to the activity of ~20–50 receptor neurons. Low concentrations of odorants yield weak spiking activity in the olfactory receptor neuron input[60]. Processing in the olfactory bulb and input from higher centers must convert the relatively small inputs into signals that are used by higher centers for odor recognition. This is supported by evidence that relatively small input signals can still evoke substantial output signals in *Drosophila*[42], and that very low odor concentrations that did not evoke clear calcium signals in the olfactory receptor neuron input were sufficient to enable odorant recognition in rats[10].

We do not know whether the output signals only reflect the action potential spiking activity that is transmitted to the cortex, or whether it reflects a combination of synaptic and spiking activity. Mitral and tufted cell action potentials can be initiated in the cell body or apical dendrites[61], and propagate throughout the cell[62]. Action potentials evoke substantially larger calcium influxes in both the cell body and dendritic tuft than do subthreshold voltage changes[63, 64]. Furthermore depolarizing synaptic activity is the main driver for action potentials and thus subthreshold signals and action potentials are likely to have a similar concentration dependence. In principle, concern about the origin of the optical signals could be mitigated in the future by using a yet to be developed protein voltage sensor whose range of voltage sensitivity does not include subthreshold potentials.

Comparison of Fig. 5 vs. Fig. 2 indicates that the output maps are more highly correlated at different concentrations of one odorant than are the maps of two different odorants. However, this result is likely dependent on the selected odorant pair. Two odorants that evoke similar activation patterns could be more highly correlated than the same odorant across concentrations.

Our finding that the olfactory bulb participates in generating the perception of concentration invariance of odor identity may not be true of all vertebrates. In the box turtle, *Terepene triunguis*, the maps of input to the bulb were already somewhat concentration invariant[65]. Additional measurements would be needed to understand the difference between mouse, zebrafish, *Drosophila* and turtle. These might guide understanding of the mechanistic principles behind generating concentration invariant maps of odor identity.

The use of ketamine/xylazine could influence the bulb's input-output relationship due to its action on NMDA receptors. Odor-evoked responses can be more variable and dynamic in awake vs. anesthetized animals[21, 35, 66]. Future studies are needed to explore the bulb's input-output transformation in unanesthetized preparations.

Our approach to performing input and output measurements in the same glomeruli should be useful in determining whether the bulb participates in other odorant perceptions. Knowing the functions carried out by the olfactory bulb will guide investigations into the function of its complex synaptic network. Furthermore, methods similar to those used here could be used to determine the input-output transformation in other brain regions as well.

## Methods

**Surgery and imaging in adult mice**. All experiments were performed in accordance with relevant guidelines and regulations, including a protocol approved by the Institutional Animal Care and Use Committees at Yale University. Protocadherin 21-Cre (Pcdh21-Cre)[26] embryos were acquired from RIKEN BioResource Center (No. RBRC02189) and recovered by the Yale Genome Editing

Center. Thy1-GCaMP6f GP5.11 transgenic mice were acquired from Jax (Stock #024339)[27]. OMP-GCaMP6f transgenic mice were generated by crossing OMP-Cre transgenic mice (Jax Stock #006668)[67] to a floxed GCaMP6f reporter transgenic mouse (Jax Stock #024105)[68]. Pcdh21-Cre and Thy1-GCaMP6f mouse used in the study expressed cre recombinase or GCaMP as determined via genotyping by Transnetyx (Cordova, TN). The olfactory bulbs were histologically processed in a subset of the OMP-GCaMP6f ($N = 4$) and Thy1-GCaMP6f ($N = 4$) transgenic mice, and visual inspection confirmed expression was located in the expected locations (Supplementary Fig. 1).

For all surgical procedures, male or female adult (40 – 100 days old) mice were anesthetized with a mixture of ketamine (90 mg kg$^{-1}$) and xylazine (10 mg kg$^{-1}$). Anesthesia was supplemented as needed to maintain areflexia, and anesthetic depth was monitored periodically via the pedal reflex. Animal body temperature was maintained at approximately 37.5 °C using a heating pad placed underneath the animal. For recovery manipulations, animals were maintained on the heating pad until awakening. Local anesthetic (1% bupivacaine, McKesson Medical) was applied to all incisions. Respiration was recorded with a piezoelectric sensor.

For virus injections animals were anesthetized, and a small hole (<1 mm) was made in the skull directly above one olfactory bulb. Adeno-associated viral (AAV) vectors were acquired from the Penn Vector Core. Cre-dependent AAV1s expressing either ArcLight (AAV1.Syn.OptiArcLightQ239GE.WPRE.SV40(#75; Cre-ON), Lot #V3335TI-S), GCaMP6f (AAV1.Syn.Flex.GCaMP6f.WPRE.SV40, Lot# CS0530), GCaMP6s (AAV1.Syn.Flex.GCaMP6s.WPRE.SV40, Lot #CS0642), or jRGECO1a (AAV1.Syn.Flex.NES-jRGECO1a.WPRE.SV40, Lot #V5041MI-S) were injected using a glass capillary (tip diameter 8–15 μm) approximately 500 μm below the surface of the bulb using a Nanoliter 2000 injector (WPI Inc., Sarasota, FL). Virus titers for ArcLight, GCaMP6f, GCaMP6s and jRGECO1a were 5.1e12 genome copies (GC) ml$^{-1}$, 1.6e13, 1.4e13, and 1.6e13 GC ml$^{-1}$, respectively. In some preparations the virus was diluted with sterile saline between 2 and 16-fold to reduce pipette tip clogging and to vary the expression levels of the sensor. The injection volumes were ~1 μl. The results from preparations with injections using different virus dilutions were similar. Mice received supplemental injections of Carprofen (rimadyl) for a minimum of 3 days after virus injection. We allowed a minimum of 14 days for sensor expression prior to optical measurements.

The venerable method described by Wachowiak and Cohen (2001) was used for nose loading of calcium dye into mice. Mice were anesthetized, placed on their back, and an 8 μl mixture of 8%/0.2% calcium dye/Triton-X was drawn into a Hamilton syringe with a flexible plastic tip, which was inserted ~10 mm into the nasal cavity. 2 μl of the dye/triton mixture was infused into the nose every 3 min. Mice were allowed to recover for at least 4 days prior to optical measurements. The organic calcium dyes Fura dextran (F-3029), Calcium Green-1 dextran (C-3713) or Oregon Green 488 BAPTA-1 dextran (O-6798) were from ThermoFisher Scientific (Waltham, MA), and Cal-590 Dextran (#20509) was acquired from AAT Bioquest (Sunnyvale, CA).

For wide-field fluorescence imaging, mice were anesthetized, and the bone above one or both olfactory bulbs was either thinned or removed. The exposure was covered with agarose and sealed with a glass coverslip. The dorsal surface of one or both hemispheres was illuminated with the appropriate excitation wavelength using epifluorescence illumination on an antediluvian Leitz Ortholux II microscope with a 150 W Xenon arc lamp (Opti Quip) and an appropriate long-pass dichroic mirror. Experiments with Fura dextran used either 340/10 nm (Chroma ET340x) or 380/10 nm (Chroma, ET380x) excitation light, a 400 nm long pass dichroic mirror, and a 510/84 nm emission filter (Semrock FF01-510/84). Oregon Green 488 BAPTA-1 dextran, Calcium Green-1 dextran, ArcLight, GCaMP6f and GCaMP6s were measured using 479 nm excitation light (Semrock FF01-479/40), a 515 nm long pass dichroic mirror, and a 530 nm long pass or 534/42 nm band pass emission filter (Semrock FF01-534/42). jRGECO1a was measured using 535/40 nm (Chroma D535/40 m) or 565/24 nm (Semrock FF01-565/24) excitation light, a 590 nm dichroic mirror, and a 610 nm long pass emission filter (Thorlabs FGL590M).

Wide-field optical signals were measured using a Nikon 10x, 0.5 NA (2.3 × 2.3 mm field of view), Nikon 16 × 0.8 NA (1.4 × 1.4 mm field of view), or Olympus 20 × 1.0 NA (0.9 × 0.9 mm field of view) objectives with a 200 mm focal length lens inserted into the emission pathway for single bulb measurements. A 4 × 0.16 NA (3.5 × 3.5 mm field of view) was used for dual bulb measurements without the lens inserted into the emission pathway.

A modified Macroscope-IIA (RedshirtImaging Inc) was used for imaging the OMP-GCaMP6f and Thy1-GCaMP6f preparations. In these experiments the light from a Prizmatix LED (UHP-T-LED-White-High-CRI) was used with a 35 mm (F/1.4) Computar CCTV lens or Nikon 10x, 0.5 NA lens. In OMP-GCaMP6f and Thy1-GCaMP6f preparations containing Cal-590 dextran dye loaded into one bulb, GCaMP6f was excited using 479 nm light and Cal-590 dextran using 572/23 nm light (Chroma ET572/23 m). A dual-band dichroic (Chroma 59009bs, transmission peaks between 510–540 nm and 590–680 nm) and emission filter (Chroma 59009 m, transmission peaks between 505–535 nm and 590–690 nm) were used in these experiments. The emission light passed through a 175 mm focal length lens.

In all experiments fluorescence emission was recorded with a NeuroCCD-SM256 camera with 2 × 2 binning between 25–125 Hz using NeuroPlex software (RedshirtImaging, Decatur, GA).

**Data analysis**. Input and output sensors were targeted to the same hemibulb in one experimental configuration (Figs 1 and 2, Supplementary Figs 2–4, $N = 5$ preparations). In these experiments, input and output were measured in alternating series of trials for each odorant concentration. The filter combination was manually switched after each odor concentration (e.g., 4 input trials using 11%, then switch filters, then 4 output trials using 11%). Fura dextran measurements were performed using both 340 nm and 380 nm excitation light for the preparation in Fig. 2. Imaging at 340 nm yielded much dimmer fluorescence, but similar results were obtained using both wavelengths. In a second experimental configuration, input and output sensors with similar spectra were targeted to opposite hemibulbs and were measured simultaneously ($N = 3$ preparations). In a third experimental configuration, input and output sensors were measured in opposite hemibulbs of the same preparation, but in separate trials ($N = 6$ preparations). Summary data from opposite bulb preparations are included in Fig. 3 and Supplementary Table 2. The input and output sensor combinations included in the population summary in Fig. 3b are shown in Supplementary Table 2. No methods of randomization were used in this study and the authors were not blind to any experimental conditions.

Odorant-evoked signals were collected in consecutive (3–20) odorant presentations separated by a minimum of 45 s. The individual trials were manually inspected, and occasional trials with obvious artifact were discarded. The onset of inhalation was defined as the first downward deflection following the first peak in the respiration recording after the start of the odorant presentation. Trials were averaged after an alignment procedure where the time of the first inspiration following the odorant presentation was identified in each trial and the recordings synchronized to this time. ArcLight signals representing depolarization, and organic calcium dye and protein signals representing calcium increases are shown upwards.

Noisy pixels receiving light from outside of the bulb, and those adjacent to major blood vessels were omitted from analysis. Data from each pixel were low-pass Gaussian filtered at 1 Hz, and had an exponential drift subtracted that was calculated based on the signal prior to the stimulus onset. The exponential fitting procedure sometimes led to a small number of pixels (~0–50 per map) with a poor fit. Those pixels were assigned an average of the neighboring 4 pixels (i.e., "fudged"). Finally, the signal from each pixel was divided by its resting fluorescence measured at the beginning of each trial.

The activation frame subtraction maps were generated by subtracting the temporal average of the 1–2 s preceding the stimulus from a 1 s temporal average around the response peak using Frame Subtraction in NeuroPlex (Fig. 2a and Supplementary Figs 2a–4a). No spatial filtering was performed except for the maps in Supplementary Fig. 2, although the activity maps were depixelated for display. Each output response map was scaled to their minimum and maximum pixel values. The input maps were scaled so that the maximum intensity value was proportional to the normalized value of the output map for that odorant concentration (e.g., in Fig. 2a, the maximum value for the 1.83% saturated vapor output map was 81.1% of the 11% saturated vapor condition. The 1.83% saturated vapor input map was scaled so that the maximum intensity value was 81.1% of its own max evoked at 11% saturated vapor).

Individual glomeruli were visually identified as "circular" peaks of activation ~50–100 μm in diameter. For same bulb preparations, glomeruli were selected if they were present for both input and output. For opposite bulb preparations, all input or output glomeruli that were activated by 1.83% saturated vapor were counted. The pixels in each ROI (Fig. 2a, Supplementary Figs 2a–4a, ROIs) were averaged to generate optical traces from each glomerulus (Figs 1c and 2c, and Supplementary Figs 2c–4c). Response amplitudes for each identified glomerulus were identified as the difference in the temporal average of the 1–2 s preceding the stimulus from a 0.8–1 s average around the peak of the signal (e.g., black bars under the 6th input trace in Fig. 2c). Glomeruli without a clearly evoked signal were assigned a value of 0 for that odorant concentration. Amplitude measurements from each glomerulus were normalized to the signal size evoked by 11% of saturated vapor.

For the data in Fig. 3b, a repeated-measures ANOVA (SPSS) was used to confirm that there was a significant effect of input-output and odor concentration. The statistical threshold was adjusted for multiple comparisons to $p = 0.016$ and Wilcoxon rank sum tests (ranksum function in MATLAB) were used to compare input and output normalized signal size at 1.83%, 0.36 and 0.12% of saturated vapor. Similar statistical significance was measured using paired t-tests, and the nonparametric Kolmogorov-Smirnov test (kstest2 function in MATLAB). Error bars in all figures indicate Standard Error of the Mean. Hill coefficients for the input and output concentration response curves were estimated using the equation $R = \frac{C^n}{C^n + k^n}$ where $R$ is the normalized response amplitude at a particular odorant concentration $C$, and $k$ is the half-saturating concentration.

Spatial correlations of the frame subtraction maps (Fig. 2b, Supplementary Figs 2b–4b) were calculated in MATLAB (Mathworks, Natick, MA) as the correlation coefficient of each map with the maps evoked by the other odorant concentrations (Fig. 3a and Supplementary Table 1a). The correlation analysis was also performed on only the subsets of pixels overlaying the glomerular ROIs identified in each preparation's activity map using unfiltered activity maps (Supplementary Table 1b).

Statistical significance (Fig. 3a and Supplementary Table 1) was measured with a Wilcoxon rank sum test using the individual correlation values for the input and output, and for comparisons between filtered and unfiltered output at each odor

condition for 13 preparations. We examined the effect of high-pass spatial filtering on the output map correlations by applying a high-pass spatial filter (41 × 41 pixel kernel) to the output maps before measuring the correlation coefficient restricted to the glomerular regions of interest. A Wilcoxon rank sum test was used to compare the unfiltered and filtered correlation coefficients. A Kruskall-Wallis Test and Mann-Whitney U-Test were used for the independent sample correlation comparisons in Fig. 4b.

A subset of the population data was corrected for a diffuse output signal by subtracting the $\Delta F/F$ measured from the area of the bulb not containing glomerular peaks of activity from the $\Delta F/F$ measured from each glomerulus at each odor concentration (Supplementary Fig. 5a, b). Cases where this subtraction yielded negative values were assigned 0. The corrected values were normalized to the corrected 11% of saturated vapor values (Supplementary Fig. 5c). This correction was performed for the input and output measurements in 7 preparations included in Supplementary Table 2).

**Odorant stimuli and delivery.** Odorants (Sigma-Aldrich) were diluted from saturated vapor with cleaned air using a flow dilution olfactometer described previously[43]. The olfactometer was designed to provide a constant flow of air blown over the nares. Odorants were constantly injected into the olfactometer, but sucked away via a vacuum that was switched off during odorant presentation. Cross contamination was avoided by using separate Teflon tubing for each odorant. Odorants were typically delivered at different concentrations between 0.12 and 11% of saturated vapor, although the odorant methyl valerate was also delivered at 0.04% in one preparation (Supplementary Fig. 2), and ~16% in another (Supplementary Fig. 3). The odorants ethyl tiglate (Fig. 1), methyl valerate (Fig. 2 and Supplementary Figs 1–3) or isoamyl acetate were used for the experiments measuring both input and output in the same preparation (details for each measurement are included in Supplementary Table 2). Methyl valerate, 2-heptanone, and isoamyl acetate were used for the data in Fig. 5 that compared the output maps evoked by different odorants. A photo-ionization detector (Aurora Scientific, Aurora, ON) was used to confirm the time course and relative concentrations presented with our olfactometer. The PID was placed next to the mouse's nose during the optical recordings.

**Histological methods.** In a subset of imaging preparations, mice were given an overdose of euthasol, decapitated, and their brains were dissected and left in 4% paraformaldehyde for a minimum of 3 days. Each olfactory bulb was embedded in 3% agarose, and cut on a vibratome in 50–100 μm thick coronal sections. Mounted sections were coverslipped with VECTASHIELD Mounting Medium with DAPI (Vector Labs, H-1500) or Propidium Iodide (Vector Labs, H-1300). Slides were examined using either a widefield epifluoresence microscope, or a Zeiss LSM-780 confocal microscope (Carl Zeiss Microsystems). Appropriate anatomical targeting of organic calcium dyes or GCaMP6f to olfactory receptor neuron input was confirmed in 12 preparations. Appropriate genetic targeting of ArcLight, GCaMP6f, and jRGECO1a to mitral and tufted cells in Pcdh21-Cre and Thy1-GCaMP6f transgenic mice was histologically confirmed in 35 preparations. The histological image in Fig. 1b is from a Pcdh21-Cre transgenic mouse in which the olfactory receptor neurons were labeled with Cal-590 dextran and the mitral and tufted output cells were labeled with an AAV-ArcLight injection. The fluorescence for the images in Fig. 1b and Supplementary Fig. 1 were detectable without additional amplification steps, were contrast-enhanced and sharpened (unsharp mask, both applied uniformly to entire image), and were cropped and pseudo-colored using Zen Lite 2011 (Carl Zeiss Microsystems), Adobe Photoshop and Illustrator (Adobe Systems Inc.).

**Data availability.** The data that support the findings of this study are available from the corresponding authors upon request.

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

## Acknowledgements

We are grateful to U. Sung for generating the plasmid used in the cre-dependent ArcLight AAV, to L. Looger, J. Akerboom, D. Kim, and the GENIE project at the Janelia Farm Research Campus for providing the GCaMP6f, GCaMP6s, and jRGECO1a expressing AAVs through the Penn Vector Core, and to J. Verhagen for providing the OMP-GCaMP6f transgenic mice. We also thank W. Ross, D. Zecevic, B. Salzberg, J. Verhagen, M. Zochowski, G. Castellucci, O. Braubach, J. Weng and Y. Choi for helpful comments on the manuscript. Supported by US NIH DC005259, DC016133 and U01 NS099691, KIST Institutional Program Multiscale Functional Connectomics 2E267190, a James Hudson Brown–Alexander Brown Coxe fellowship from Yale University, and a Ruth L. Kirschstein National Research Service Award DC012981. D.A.S. and L.B.C. conceived the experiments, analyzed the data, created the figures and wrote the manuscript. D.A.S. performed the experiments.

## Additional information

**Competing interests:** The authors declare no competing financial interests.

