## [Peer Review File · Nature Communications]

Reviewers' comments:

Reviewer #1 (Remarks to the Author):

This paper compares odor representations at the level of primary sensory input to the olfactory bulb with those measured from mitral/tufted cell output neurons from the bulb in order to infer the function of olfactory bulb circuits in the context of odor intensity versus odor quality coding. The authors use imaging with synthetic calcium-sensitive dyes venerably loaded into the sensory neurons versus genetically-encoded calcium or voltage reporters expressed in mitral/tufted cells, and conclude based on comparisons of response patterns that representations at the output neuron level are more concentration-invariant than those at the sensory neuron input level. This topic is fundamental to understanding sensory systems in general, and of longstanding interest in the olfactory neurobiology community. However, this paper as it stands does not do an adequate job of addressing the issue: this is essentially one type of experiment with some fundamental limitations in data interpretation, to the point that I am not even confident that the differences in concentration-dependence reflect technical aspects of the measurements or actual differences in odor coding by the input and output neurons. Explanation of these concerns are below.

1. The conclusions rely on comparing how the linear correlation between response patterns changes as odor concentration is varied, for sensory neurons and mitral/tufted cells. However, sensory neurons are only tested using a synthetic dye (fura dextran), while mitral/tufted cells are tested using GCaMPs (or, in some cases, ArcLight), which have a different affinity for calcium, very different signal-to-noise ratio, different dynamic range and different kinetics. Any one of these differences could account for the higher correlation across concentration that is seen for the 'output signal' than for the input signal. In addition, the input signal is reporting calcium in the axon terminals of the sensory neurons, while the 'output' signal is reporting (presumably) calcium in the dendrites of mitral/tufted cells, which in itself could differentially report neural activity. Additional control experiments (such as, for starters, expressing GCaMPs or rGECO in the sensory neurons) are important to understand the degree to which these differences might account for the observed results.
2. In nearly every example shown, and for the summary plots shown (i.e., Fig 2D, 3B, etc.), the 'Output' signal never reaches zero or near-zero. Thus, the experimenters have not covered the full dynamic range of response for the mitral cells and may in fact only be sampling the upper half. Thus, the output signal could well be highly concentration-variant, but the concentration-response function could just be shifted to the left due to higher sensitivity of the mitral cells (or the reporter), and the experimenters have only sampled the part of the range where responses are less concentration-dependent.
3. Similarly, in the examples shown - such as in Figure 2C, it appears that every glomerulus shows a substantial output signal even at the lowest concentration tested. Why is this? More to the point, given that these signals are all measured with epifluorescence, there is likely a substantial contribution of signal from mitral/tufted cell lateral dendrites, which will lead to a broadly-distributed and non-selective signal - which would also lead to a higher correlation coefficient for response patterns. Thus it seems quite possible that the result on which the main conclusion is based could be explained solely by this technical issue. This should be addressed by imaging with non-antediluvian two-photon microscopy to

isolate signal arising from the glomerulus and exclude lateral dendrite signals.

4. I am curious as to why, with the ArcLight imaging experiments, the odor-evoked signal appears to have the same relatively slow temporal dynamics as the signal imaged with GCaMPs.

Reviewer #2 (Remarks to the Author):

This manuscript addresses a narrowly defined, yet interesting and fundamental, question in olfactory neuroscience that is of broader relevance. The question whether the olfactory bulb participates in the generation of concentration-invariant representations of odors has, to some extent, been addressed by a few previous studies with similar conclusions. Nevertheless, this paper adds important and convincing evidence that the olfactory bulb is indeed involved in concentration invariance computations. The authors address this question very directly by an elegant two-color activity imaging approach. The results are clear and convincing. This paper is therefore an important contribution that provides a clear answer to a straightforward fundamental question. The paper is also interesting from a technical point of view. Although the manuscript is rather short and based only on one type of experiment, I feel that it can, in principle, fulfill the standards for publication because the results are convincing and important. Nevertheless, the manuscript should still be improved, particularly the discussion of the authors' results in the context of previous studies.

Specific comments:

1. Abstract: the authors state that "...this is the first direct evidence that the mammalian olfactory bulb participates in generating the perception of concentration invariance of odor quality...". I feel this is an overstatement. There have been at least two recent papers providing strong evidence that the olfactory bulb is involved in generating concentration-invariant odor representations. One is Zhu et al (Nature Neuroscience 2013; this is zebrafish, not mammalian, but still highly relevant). This paper demonstrates directly that the olfactory bulb enhances concentration invariance of mitral cell activity and pinpoints the mechanism. It should be cited and discussed. The other paper is Banerjee et al (Neuron 2015), also highly relevant. This is cited, but not really acknowledged in the introduction for implicating the olfactory bulb in concentration invariance. I believe the problem here is that the authors tacitly imply that the generation of concentration invariance can be studied only by measuring inputs and outputs as a function of concentration in the same bulbs. But there are also many other ways to address this question.

2. Fig 2b and similar figures: why is there no black bar with $r=1$ at 1.83% vapor concentration?

3. Figs 2b, 3a and related Supplementary Figs: The authors may want to consider showing all pairwise correlations, not only the correlation to one reference (1.83%).

4. The authors "...conclude that the input-output transformation of the olfactory bulb

contributes to the perception that the quality of an odorant is considered the same over a range of concentrations". I feel that this statement is too strong. Their results clearly demonstrate that output responses are more concentration-invariant than input responses, and it is indeed very likely that this concentration-invariance of responses contributes to the concentration-invariance of perception. However, there is no direct proof that this transformation is causally involved in the concentration-invariance of perception. Formal "proof" requires some kind of manipulation that selectively disrupts concentration-invariance of output responses. This manipulation should then also disrupt concentration-invariance of perception. However, as it is currently unclear how such a manipulation could be achieved, this experiment is way beyond the scope of this paper. Even in the absence of such an experiment, the paper presents compelling evidence for the hypothesis that the olfactory bulb is involved in the generation of concentration-invariant odor responses. I therefore feel that conclusion should be softened somewhat. The results are nevertheless strong and interesting.

5. The authors state that "Wachowiak and Cohen (2001) and Bozza et al. (2004) found that the glomerular maps of input to the olfactory bulb were a confound of odorant identity and odorant concentration". While technically correct I feel that the specific notion of these two studies does not do justice to a number of other studies that arrived at the same conclusion. Similar imaging experiments in the olfactory bulb/antennal lobe were done previously (and subsequently) in insects and zebrafish. Although these are not mammals, they nevertheless deserve credit, in my opinion. The same conclusion had also been reached based on single-unit recordings from olfactory receptor neurons in a wide range of species.

6. In the discussion the authors state that the only other vertebrate in which the generation of "...the perception of concentration invariance..." has been studied is the box turtle. This is not true. As mentioned above, concentration invariance has been studied also in the olfactory bulb of zebrafish (Zhu et al., Nature Neurosci 2013), and a large amount of work in insects (see work by Rachel Wilson and many others) is also relevant.

7. "...The venerable method...", "...antediluvian..."

Reviewer #3 (Remarks to the Author):

This study uses wide-field epifluorescence imaging to examine the input-output relationship of olfactory bulb glomeruli in response to changes in odor concentration. The authors probe olfactory sensory neuron (OSN) glomerular input via intranasal loading of calcium dyes and measure output using viral expression of genetically encoded calcium or voltage indicators in postsynaptic mitral and tufted (MT) cells. By taking advantage of differences in the spectral properties of the indicators, the authors can study pre- and postsynaptic responses from the same glomeruli. Results suggest that patterns of glomerular output are relatively constant over a range of odor concentrations even though input maps change markedly under the same conditions. The authors propose that this provides direct evidence that the olfactory bulb participates in generating the perception of concentration invariance of odor

quality.

This manuscript uses a clever approach to image pre- and postsynaptic activity and the results are presented in a very clear and concise manner. However, there are technical concerns that should be addressed to support claims made in the study. First, although the authors do a nice job of testing a range of calcium dyes in OSNs and genetically encoded indicators in M/T cells, it is not clear that glomerular imaging actually provides a true measure of the olfactory bulb input-output relationship under their conditions. Although calcium signals derived from OSN nerve terminals are presumed to be directly proportional to odor-evoked action potentials (APs), this is unlikely to be the case for wide-field glomerular imaging of M/T dendritic tuft calcium signals. Indeed, previous studies (i.e. Charpak et al, 2001, Kato et al 2012) have shown that calcium indicators in M/T glomerular dendrites can report activity that is independent of APs (subthreshold responses). One possibility is that the "concentration invariance" of output maps is reflecting the sensitivity of indicators to local dendritic synaptic activity rather than back propagating APs. Although the authors also use the voltage indicator ArcLight, my understanding is that this sensor is also sensitive to subthreshold changes in membrane potential. Two-photon ensemble (calcium) imaging of mitral cell bodies--which better reflect AP output--could be used to support the hypothesis that odor identity is conserved during large changes in odor concentration.

Another major concern reflects the fact that all experiments are done in anesthetized animals. A number of previous studies have shown that odor-evoked responses are much more variable and dynamic in awake vs. anesthetized animals (Kato et al 2012, Wachowiak et al 2013, Sirotin et al 2015). Given the claims the authors make regarding "odor perception" I think it would be particularly important to know the input-output transformation in experiments on awake, head-fixed mice.

Reviewer #4 (Remarks to the Author):

The goal of this project was to compare the neural inputs and outputs of the olfactory bulb to assess whether the bulb might transform the neural representation of odors to make them invariant across odor concentrations. The authors used a clever approach to accomplish this, introducing short-wavelength calcium indicators into the olfactory nerve terminals via an anterograde tracer while simultaneously introducing longer-wavelength genetically encoded calcium or voltage indicators into the olfactory bulb's mitral cells. In some cases they were able to observe the response to a series of odor concentrations in both neuronal populations in the same olfactory bulb, while in others they compared the signals in the (roughly symmetrical) left and right bulbs of the same animal to confirm there was no interaction between the indicators. The authors indeed found that the activity of the mitral cells was less dependent on odor concentration than their inputs were, suggesting that a key function of the olfactory bulb circuit is to transform the peripheral input into a representation of odor identity. The results of this paper are potentially very important, both

because they inform the classic question of how the olfactory system untangles odor identity and concentration and more broadly because it shows a powerful way to isolate the computational function implemented by a neural structure. These are difficult experiments that are performed well. However, the manuscript has significant but surmountable flaws, including interpretational challenges, overly narrow analyses of the data, and statistical errors.

Major Comments

The largest concern in any experiment of this type is the technical challenge of separating limitations in the dynamic range of the optical indicators from the actual dynamic range of the underlying neural circuit. The authors make a deliberate effort to contend with this challenge, principally by showing similar results while using seven different combinations of indicators with similar distributions of K_d 's (calcium dissociation constant) in the inputs and outputs. Unfortunately, a contribution of indicator limitations nonetheless cannot be entirely ruled out. The genetically-encoded calcium indicators used in the mitral cells inevitably have higher cooperativity (Hill coefficients) than the synthetic dyes used in the olfactory nerve. There are also differences in sensitivity, such as in Fig. 2C2 and Fig. S1C1, where the lowest concentration of odor evoked no demonstrable signal in the ORN input but nonetheless evoked a large signal in the mitral cells (the authors do not address this point). Even when the indicator is the same, differences in expression or loading can produce different saturation kinetics from animal to animal - for instance comparing the Fura signals in their table shows that the ORN response to 1.83% s.v. methyl valerate was 93% of maximum for preparation 2 (e.g. near saturation) and only 42% of maximum for preparation 4 (not at all saturated) - and that variance could reflect dye loading differences or neural activity differences. As a reviewer I am torn by these data, because the authors have used just about every available method and I am convinced that their main result is qualitatively valid, but the confounding effect of indicator makes it difficult to judge effect size and reliability. The ideal experiment would be to put the same indicator into the input and the output of the bulb and compare, but this is technically difficult because ORNs seem to be impossible to transfect with AAVs. If available, the authors should consider trying an OMP-cre driver line crossed with a GCaMP6-floxed-STOP reporter line, so that they could compare the input and output using the same indicator in both populations, albeit in separate mice.

The authors do not analyze the temporal element of their data at all, which seems like a considerable oversight in a paper that aspires to assess the bulb's input-output function. While differences in indicator kinetics and the underlying biological variable being measured might make it hard to interpret peak latencies between input and output, the authors should at least report risetimes and latencies. Moreover, there seem to be notable instances of latency changes as a function of odor concentration (e.g. Fig. S1C) that occur within the same experiment, and these should be explored.

Throughout the paper the authors refer to the mitral and tufted cell signal as the "output" of the olfactory bulb. However, it is unclear (and perhaps unknowable) how much of their signal reflects mitral cell firing (e.g. actual bulb output) and how much reflects dendritic calcium signaling (e.g. local processing of input). If anything this suggests that the shift toward concentration invariance is underestimated by their data. This issue bears textual

consideration and clearer language.

The authors do a commendable job of showing all of their data, and they generally show clear effects, but the data deserve more than a perfunctory statistical analysis. The statistical analysis they do provide is poorly described and in some places definitely wrong. It is not appropriate to use parametric tests like a t-test on correlation coefficients because these values cannot be normally distributed. Use a non-parametric test like the Mann-Whitney U-test for such comparisons. It is unclear throughout exactly what statistics were being calculated, but it appeared that in some places there were repeated t-tests performed when the experimental design called for a repeated measures ANOVA (or perhaps the non-parametric version called Friedman's ANOVA) followed by post-hoc testing. For the concentration-response functions, the authors should also consider whether they would be better off performing curve fits and comparing the parameters of those fits instead of using concentration by concentration comparisons. Please report actual values for statistics, including degrees of freedom.

In the methods it is noted that glomeruli were included in the analysis only if they were present in both the input and the output maps. That seems potentially troubling if part of the input-output transformation is gating the input. Please clarify how many glomeruli were excluded and whether this could influence the analysis.

Minor Comments

The authors frequently use imprecise terminology in discussing their results that effectively exaggerates their claims. For instance, their data shows increased concentration invariance of neural odor representations but certainly do not provide "direct evidence" that the bulb contributes to the "perception" of concentration invariance. Similarly, in the discussion the authors state that "our results show that odorant identity is more likely determined by the glomerular output of the olfactory bulb," but the present experiments and analyses did not actually compare odor identity representations by comparing different odors. The discussion paragraph about the number of neurons activated seems speculative and inconsistent with the seeming insensitivity of Fura.

The writing of the manuscript needs improvement. The introduction does not adequately consider the existing literature on concentration coding in the olfactory system (consider Stopfer et al. 2003 Identity vs concentration coding in an olfactory system). The language of the introduction also is frequently vague (e.g. "a long-standing question in neuroscience") and constantly makes reference to "maps," a term of art among imagers, when it would more effectively refer to spatial patterns of neural activity or patterns of sensory input across glomeruli, etc. It seems weird to refer to the retina in the abstract and nowhere else. It would be good to discuss the computational utility of concentration invariant representations for downstream regions.

The use of ketamine is tricky for this experiment because it is an NMDAR antagonist and thus could be influencing the input-output relationship. However, it is unclear if NMDARs play a major role in this synapse and alternatives like pentobarbital and isoflurane have their own, possibly more severe problems. I think there should be a caveat in the discussion

about this issue.

The scaling of the response maps needs to be made easier for an inexperienced reader to follow, and a grayscale scale bar is needed. Numbering the regions of interest in the maps would help match up the traces.

Somewhere specify that the error bars on graphs refer to standard errors.

Fig. 2b does not match its caption.

If the "antediluvian" microscope has special optics that enable UV imaging of the venerable fura indicator, the methods should indicate that.

We are grateful for the thoughtful and helpful comments provided by each reviewer.

Reviewers' comments

Reviewer #1 (Remarks to the Author):

This paper compares odor representations at the level of primary sensory input to the olfactory bulb with those measured from mitral/tufted cell output neurons from the bulb in order to infer the function of olfactory bulb circuits in the context of odor intensity versus odor quality coding. The authors use imaging with synthetic calcium-sensitive dyes venerably loaded into the sensory neurons versus genetically-encoded calcium or voltage reporters expressed in mitral/tufted cells, and conclude based on comparisons of response patterns that representations at the output neuron level are more concentration-invariant than those at the sensory neuron input level. This topic is fundamental to understanding sensory systems in general, and of longstanding interest in the olfactory neurobiology community. However, this paper as it stands does not do an adequate job of addressing the issue: this is essentially one type of experiment with some fundamental limitations in data interpretation, to the point that I am not even confident that the differences in concentration-dependence reflect technical aspects of the measurements or actual differences in odor coding by the input and output neurons. Explanation of these concerns are below.

1. The conclusions rely on comparing how the linear correlation between response patterns changes as odor concentration is varied, for sensory neurons and mitral/tufted cells. However, sensory neurons are only tested using a synthetic dye (fura dextran), while mitral/tufted cells are tested using GCaMPs (or, in some cases, ArcLight), which have a different affinity for calcium, very different signal-to-noise ratio, different dynamic range and different kinetics. Any one of these differences could account for the higher correlation across concentration that is seen for the 'output signal' than for the input signal. In addition, the input signal is reporting calcium in the axon terminals of the sensory neurons, while the 'output' signal is reporting (presumably) calcium in the dendrites of mitral/tufted cells, which in itself could differentially report neural activity. Additional control experiments (such as, for starters, expressing GCaMPs or rGECO in the sensory neurons) are important to understand the degree to which these differences might account for the observed results.

Good comment. We were also concerned about the potential differences in indicator biophysics. This is why we had used several different organic calcium dyes and protein sensors. We added several new experiments that we think directly address this concern.

First, we added the proposed experiment where we repeated the input measurements using a transgenic mouse that expresses GCaMP6f in the olfactory receptor neurons. We compared the measurements from these mice to preparations in which output measurements were performed using GCaMP6f in both the Pcdh21-Cre transgenic mice (GCaMP6f virally expressed), and in a second transgenic mouse that selectively expresses GCaMP6f in bulb M/T cells (Thy1-GCaMP6f). The results were similar to the results from the same-hemibulb using different sensors, and are included in a new figure (**Fig. 4**).

Second, we compared the input activity maps measured using an organic calcium dye and GCaMP6f in the same bulb by loading the red-shifted organic calcium dye Cal-590 dextran into the olfactory epithelium in two OMP-GCaMP6f preparations. The activity maps measured using the organic dye and protein sensor were qualitatively similar, and the results are included in a new supplemental figure (**Supplementary Fig. 6**).

We believe these new data provide convincing evidence that our reported differences between input and output are not related to sensor differences. However, the point about differences in how the input axons and output dendrites report activity is also another important concern. Mitral and tufted cell action potentials can be initiated in both the cell body and dendrites and propagate throughout the cell. Action potentials result in calcium influx throughout the cell including the glomerular tufts. Although subthreshold membrane potentials do evoke calcium changes, they are considerably smaller than those evoked by action potentials (references in the manuscript). Furthermore, synaptic contamination from the input would reduce the difference between input and output. A more convincing answer to the concern would be to use a protein voltage sensor whose range of voltage sensitivity did not include synaptic potentials. At present a virus with a floxed version of such a voltage sensor does not exist. We have added in words to this effect in the discussion.

2. In nearly every example shown, and for the summary plots shown (i.e., Fig 2D, 3B, etc.), the 'Output' signal never reaches zero or near-zero. Thus, the experimenters have not covered the full dynamic range of response for the mitral cells and may in fact only be sampling the upper half. Thus, the output signal could well be highly concentration-variant, but the concentration-response function could just be shifted to the left due to higher sensitivity of the mitral cells (or the reporter), and the experimenters have only sampled the part of the range where responses are less concentration-dependent.

Perceptual concentration invariance does not always extend over a very large concentration range (Gross-Isseroff 1988). That said, we have not sampled the full dynamic range of response for the mitral cells. We have occasionally

measured output signals at a lower odor concentration (e.g., 0.04%, **Supplementary Fig. S2**), although never signals from the input. Because of this inconsistency we restricted our population analysis to a range of concentrations that usually had detectable output signals.

3. Similarly, in the examples shown - such as in Figure 2C, it appears that every glomerulus shows a substantial output signal even at the lowest concentration tested. Why is this?

The signal-to-noise ratio in the dye measurements have a threshold that is ~2-5% of the largest signal. We speculate that if the largest signal represents the activation of ~1000 olfactory receptor neurons, then the smallest detectable signal would represent the activity of ~20-50 receptor neurons. The activation of relatively few receptor neurons can elicit detectable responses in output cells in *Drosophila* (Kazama and Wilson 2008). Furthermore, concentrations of odorants similar to or much lower than those used in our current manuscript that fail to evoke detectable input signals can still be recognized by rodents (Homma et al., 2009).

Also, we note that in many cases where there is an output signal at the lowest odorant concentration, there is still a detectable glomerular peak of activation in the activity map.

More to the point, given that these signals are all measured with epifluorescence, there is likely a substantial contribution of signal from mitral/tufted cell lateral dendrites, which will lead to a broadly-distributed and non-selective signal - which would also lead to a higher correlation coefficient for response patterns. Thus it seems quite possible that the result on which the main conclusion is based could be explained solely by this technical issue. This should be addressed by imaging with non-antidiluvian two-photon microscopy to isolate signal arising from the glomerulus and exclude lateral dendrite signals.

The reviewer has brought up an important technical caveat about our experiments. Although it is likely that there are non-selective signal contributions from mitral and tufted cell lateral dendrites, this issue cannot explain the main result which is that the glomerular output maps are more consistent. That is, a non-selective signal from the lateral dendrites will not generate glomerular shaped maps of activation. That said, the concern that the non-selective signal could alter the results is reasonable, and something we had worried about. We attempted to deal with this issue using two new analyses that have been added to the paper.

1. The correlation analysis was performed again, but we only included pixels that contained the glomerular peaks of activity. Because the glomerular peaks of activity cannot be explained by lateral dendrite activity we thought this was a fair compromise. This change yielded similar differences between input and output with similar levels of statistical significance.
2. We attempted to control for the effect of the diffuse signal on the output signal sizes by measuring the diffuse signal calculated from the parts of the bulb not containing glomerular peaks of activity. The diffuse signal was subtracted from the signal sizes measured from the glomerular regions of interest. We found that this correction yielded very similar normalized results showing that the output was relatively concentration invariant. A new supplementary figure was added illustrating this analysis (**Supplementary Fig. 5**).

We think this reasoning and additional analyses are convincing. We have added words to this effect in the discussion. Repeating our experiments using 2-photon imaging is certainly appropriate but it is difficult because the Yale laboratory does not have a 2-photon microscope.

4. I am curious as to why, with the ArcLight imaging experiments, the odor-evoked signal appears to have the same relatively slow temporal dynamics as the signal imaged with GCaMPs.

There are two reasons. First, the traces shown in the manuscript were low-pass filtered at 1 Hz. Second, this measurement was performed in the rostral olfactory bulb, in which the glomerular kinetics are relatively slow (previously described by Spors et al., 2006; Wachowiak et al., 2013 and Storace et al., 2015).

Reviewer #2 (Remarks to the Author):

This manuscript addresses a narrowly defined, yet interesting and fundamental, question in olfactory neuroscience that is of broader relevance. The question whether the olfactory bulb participates in the generation of concentration-invariant representations of odors has, to some extent, been addressed by a few previous studies with similar conclusions. Nevertheless, this paper adds important and convincing evidence that the olfactory bulb is indeed involved in concentration invariance computations. The authors address this question very directly by an elegant two-color activity imaging approach. The results are clear and convincing. This paper is therefore an important contribution that provides a clear answer to a straightforward fundamental question. The paper is also interesting from a technical point of view.

Although the manuscript is rather short and based only on one type of experiment, I feel that it can, in principle, fulfill the standards for publication because the results are convincing and important. Nevertheless, the manuscript should still be improved, particularly the discussion of the authors' results in the context of previous studies.

Specific comments:

1. Abstract: the authors state that "...this is the first direct evidence that the mammalian olfactory bulb participates in generating the perception of concentration invariance of odor quality...". I feel this is an overstatement. There have been at least two recent papers providing strong evidence that the olfactory bulb is involved in generating concentration-invariant odor representations. One is Zhu et al (Nature Neuroscience 2013; this is zebrafish, not mammalian, but still highly relevant). This paper demonstrates directly that the olfactory bulb enhances concentration invariance of mitral cell activity and pinpoints the mechanism. It should be cited and discussed. The other paper is Banerjee et al (Neuron 2015), also highly relevant. This is cited, but not really acknowledged in the introduction for implicating the olfactory bulb in concentration invariance. I believe the problem here is that the authors tacitly imply that the generation of concentration invariance can be studied only by measuring inputs and outputs as a function of concentration in the same bulbs. But there are also many other ways to address this question.

Good comment. The reference and discussion of Zhu et al. was added, and we have done a better job of emphasizing the importance of Banerjee. et al.

2. Fig 2b and similar figures: why is there no black bar with $r=1$ at 1.83% vapor concentration?

3. Figs 2b, 3a and related Supplementary Figs: The authors may want to consider showing all pairwise correlations, not only the correlation to one reference (1.83%).

Good suggestion. We have replaced the bar graphs with correlation matrices. All correlations are now shown. Significance values (calculated using Wilcoxon signed-rank test) for the population matrix in **Fig. 3a** are now included in **Supplementary Table 1**.

4. The authors "...conclude that the input-output transformation of the olfactory bulb contributes to the perception that the quality of an odorant is considered the same over a range of concentrations". I feel that this statement is too strong. Their results clearly demonstrate that output responses are more concentration-invariant than input responses, and it is indeed very likely that this concentration-invariance of responses contributes to the concentration-invariance of perception. However, there is no direct

proof that this transformation is causally involved in the concentration-invariance of perception. Formal "proof" requires some kind of manipulation that selectively disrupts concentration-invariance of output responses. This manipulation should then also disrupt concentration-invariance of perception. However, as it is currently unclear how such a manipulation could be achieved, this experiment is way beyond the scope of this paper. Even in the absence of such an experiment, the paper presents compelling evidence for the hypothesis that the olfactory bulb is involved in the generation of concentration-invariant odor responses. I therefore feel that conclusion should be softened somewhat. The results are nevertheless strong and interesting.

Good criticism. We altered the wording in the discussion and added in a sentence emphasizing that proof would require a manipulation of the mechanisms underlying concentration-invariant odor perception.

5. The authors state that "Wachowiak and Cohen (2001) and Bozza et al. (2004) found that the glomerular maps of input to the olfactory bulb were a confound of odorant identity and odorant concentration". While technically correct I feel that the specific notion of these two studies does not do justice to a number of other studies that arrived at the same conclusion. Similar imaging experiments in the olfactory bulb/antennal lobe were done previously (and subsequently) in insects and zebrafish. Although these are not mammals, they nevertheless deserve credit, in my opinion. The same conclusion had also been reached based on single-unit recordings from olfactory receptor neurons in a wide range of species.

6. In the discussion the authors state that the only other vertebrate in which the generation of "...the perception of concentration invariance..." has been studied is the box turtle. This is not true. As mentioned above, concentration invariance has been studied also in the olfactory bulb of zebrafish (Zhu et al., Nature Neurosci 2013), and a large amount of work in insects (see work by Rachel Wilson and many others) is also relevant.

(Response to #5-6): The reviewer is correct. References to zebrafish, *Drosophila* and other mammalian reports have been added to the Introduction and Discussion.

7. "...The venerable method...", "...antediluvian..."

Reviewer #3 (Remarks to the Author):

This study uses wide-field epifluorescence imaging to examine the input-output relationship of olfactory bulb glomeruli in response to changes in odor concentration.

The authors probe olfactory sensory neuron (OSN) glomerular input via intranasal loading of calcium dyes and measure output using viral expression of genetically encoded calcium or voltage indicators in postsynaptic mitral and tufted (MT) cells. By taking advantage of differences in the spectral properties of the indicators, the authors can study pre- and postsynaptic responses from the same glomeruli. Results suggest that patterns of glomerular output are relatively constant over a range of odor concentrations even though input maps change markedly under the same conditions. The authors propose that this provides direct evidence that the olfactory bulb participates in generating the perception of concentration invariance of odor quality.

This manuscript uses a clever approach to image pre- and postsynaptic activity and the results are presented in a very clear and concise manner. However, there are technical concerns that should be addressed to support claims made in the study. First, although the authors do a nice job of testing a range of calcium dyes in OSNs and genetically encoded indicators in M/T cells, it is not clear that glomerular imaging actually provides a true measure of the olfactory bulb input-output relationship under their conditions. Although calcium signals derived from OSN nerve terminals are presumed to be directly proportional to odor-evoked action potentials (APs), this is unlikely to be the case for wide-field glomerular imaging of M/T dendritic tuft calcium signals. Indeed, previous studies (i.e. Charpak et al, 2001, Kato et al 2012) have shown that calcium indicators in M/T glomerular dendrites can report activity that is independent of APs (subthreshold responses). One possibility is that the "concentration invariance" of output maps is reflecting the sensitivity of indicators to local dendritic synaptic activity rather than back propagating APs. Although the authors also use the voltage indicator ArcLight, my understanding is that this sensor is also sensitive to subthreshold changes in membrane potential.

This is an important concern. A more convincing answer to the concern would be to use a protein voltage sensor whose range of voltage sensitivity did not include synaptic potentials. At present a virus with a floxed version of such a voltage sensor does not exist. However, any olfactory receptor neuron input synaptic contamination of the ArcLight signal would reduce the difference between input and output. Thus, we think that our measurement most likely reflects the mitral/tufted cell output. We have added words to this effect in the discussion.

Two-photon ensemble (calcium) imaging of mitral cell bodies--which better reflect AP output--could be used to support the hypothesis that odor identity is conserved during large changes in odor concentration. Another major concern reflects the fact that all experiments are done in anesthetized animals. A number of previous studies have shown that odor-evoked responses are much more variable and dynamic in awake vs. anesthetized animals (Kato et al 2012, Wachowiak et al 2013, Sirotin et al 2015). Given the claims the authors make regarding "odor perception" I think it would be particularly

important to know the input-output transformation in experiments on awake, head-fixed mice.

We agree that all of these are excellent ideas, although we hope the reviewer will understand they are beyond the scope of the present manuscript. We have now emphasized the fact that our measurements were carried out in anesthetized animals in the discussion.

Reviewer #4 (Remarks to the Author):

The goal of this project was to compare the neural inputs and outputs of the olfactory bulb to assess whether the bulb might transform the neural representation of odors to make them invariant across odor concentrations. The authors used a clever approach to accomplish this, introducing short-wavelength calcium indicators into the olfactory nerve terminals via an anterograde tracer while simultaneously introducing longer-wavelength genetically encoded calcium or voltage indicators into the olfactory bulb's mitral cells. In some cases they were able to observe the response to a series of odor concentrations in both neuronal populations in the same olfactory bulb, while in others they compared the signals in the (roughly symmetrical) left and right bulbs of the same animal to confirm there was no interaction between the indicators. The authors indeed found that the activity of the mitral cells was less dependent on odor concentration than their inputs were, suggesting that a key function of the olfactory bulb circuit is to transform the peripheral input into a representation of odor identity. The results of this paper are potentially very important, both because they inform the classic question of how the olfactory system untangles odor identity and concentration and more broadly because it shows a powerful way to isolate the computational function implemented by a neural structure. These are difficult experiments that are performed well. However, the manuscript has significant but surmountable flaws, including interpretational challenges, overly narrow analyses of the data, and statistical errors.

Major Comments

The largest concern in any experiment of this type is the technical challenge of separating limitations in the dynamic range of the optical indicators from the actual dynamic range of the underlying neural circuit. The authors make a deliberate effort to contend with this challenge, principally by showing similar results while using seven different combinations of indicators with similar distributions of K_d 's (calcium dissociation constant) in the inputs and outputs. Unfortunately, a contribution of indicator limitations nonetheless cannot be entirely ruled out. The genetically-encoded calcium indicators used in the mitral cells inevitably have higher cooperativity (Hill coefficients) than the synthetic dyes used in the olfactory nerve.

See below. Also, the higher Hill coefficients of the GCaMPs would only serve to make the output more concentration dependent (Rose et al., 2014). If anything this suggests that the output may be even more concentration invariant than what we report.

There are also differences in sensitivity, such as in Fig. 2C2 and Fig. S1C1, where the lowest concentration of odor evoked no demonstrable signal in the ORN input but nonetheless evoked a large signal in the mitral cells (the authors do not address this point).

The signal-to-noise ratio in the dye measurements has a threshold that is ~2-5% of the largest signal. We speculate that if the largest signal represents the activation of ~1000 olfactory receptor neurons, then the smallest detectable signal would represent the activity of ~20-50 receptor neurons. The activation of relatively few receptor neurons can elicit detectable responses in output cells in *Drosophila* (Kazama and Wilson 2008). Furthermore, low concentrations of odorants (similar to or much smaller than those used in our current manuscript) that fail to evoke detectable input signals can still be recognized by rats (Homma et al., 2009).

Even when the indicator is the same, differences in expression or loading can produce different saturation kinetics from animal to animal - for instance comparing the Fura signals in their table shows that the ORN response to 1.83% s.v. methyl valerate was 93% of maximum for preparation 2 (e.g. near saturation) and only 42% of maximum for preparation 4 (not at all saturated) - and that variance could reflect dye loading differences or neural activity differences.

Clearly there are animal to animal differences. However, we are uncertain about their origin.

As a reviewer I am torn by these data, because the authors have used just about every available method and I am convinced that their main result is qualitatively valid, but the confounding effect of indicator makes it difficult to judge effect size and reliability. The ideal experiment would be to put the same indicator into the input and the output of the bulb and compare, but this is technically difficult because ORNs seem to be impossible to transfect with AAVs. If available, the authors should consider trying an OMP-cre driver line crossed with a GCaMP6-floxed-STOP reporter line, so that they could compare the input and output using the same indicator in both populations, albeit in separate mice.

We were also concerned about the potential differences in indicator biophysics. This is why we had used several different organic calcium dyes and

protein sensors. We added several new experiments that we think directly address this concern.

First, we added the proposed experiment where we repeated the input measurements using a transgenic mouse that expresses GCaMP6f in the olfactory receptor neurons. We compared the measurements from these mice to preparations in which output measurements were performed using GCaMP6f in both the Pcdh21-Cre transgenic mice (GCaMP6f virally expressed), and in a second transgenic mouse that selectively expresses GCaMP6f in the M/T cells in the bulb (Thy1-GCaMP6f). The results were similar to the results from the same-hemibulb using different sensors, and are included in a new figure (**Fig. 4**).

Second, we compared the input activity maps measured using an organic calcium dye (Cal-590) and GCaMP6f in the same bulb by loading the red-shifted organic calcium dye (Cal-590 dextran) into the olfactory epithelium in two OMP-GCaMP6f preparations. The activity maps measured using the organic dye and protein sensor were qualitatively similar, and the results are included in a new supplemental figure (**Supplementary Fig. 6**).

The authors do not analyze the temporal element of their data at all, which seems like a considerable oversight in a paper that aspires to assess the bulb's input-output function. While differences in indicator kinetics and the underlying biological variable being measured might make it hard to interpret peak latencies between input and output, the authors should at least report risetimes and latencies. Moreover, there seem to be notable instances of latency changes as a function of odor concentration (e.g. Fig. S1C) that occur within the same experiment, and these should be explored.

The presently available sensors are not ideal for temporal comparisons. The calcium signals are large but relatively slow (Storage et al., 2015) and the voltage signals are small and thus the signal-to-noise ratio in high temporal resolution measurements is too low for accurate comparisons (see Storage et al., 2015 Figure 5 for an unfiltered ArcLight trace). What is needed is both red and green voltage sensors with large, fast signals. While efforts are underway to develop such sensors, it is uncertain if and when they might be available.

Throughout the paper the authors refer to the mitral and tufted cell signal as the "output" of the olfactory bulb. However, it is unclear (and perhaps unknowable) how much of their signal reflects mitral cell firing (e.g. actual bulb output) and how much reflects dendritic calcium signaling (e.g. local processing of input). If anything this suggests that the shift toward concentration invariance is underestimated by their data. This issue bears textual consideration and clearer language.

This is an important concern. Although mitral and tufted cell action potentials initiated in either the cell body or dendrites propagate throughout the cell causing calcium influx, subthreshold potentials do evoke calcium changes. However, they are considerably smaller than those evoked by action potentials (references in the manuscript). However, any olfactory receptor neuron input synaptic contamination of the ArcLight signal would reduce the difference between input and output. Thus we think that our measurement mostly reflects the mitral/tufted cell output. A more convincing answer to this concern would be to use a protein voltage sensor whose range of voltage sensitivity did not include synaptic potentials. At present a virus with a floxed version of such a voltage sensor does not exist. We have added words to the Discussion that discusses this concern.

The authors do a commendable job of showing all of their data, and they generally show clear effects, but the data deserve more than a perfunctory statistical analysis. The statistical analysis they do provide is poorly described and in some places definitely wrong. It is not appropriate to use parametric tests like a t-test on correlation coefficients because these values cannot be normally distributed. Use a non-parametric test like the Mann-Whitney U-test for such comparisons.

Reviewer #4 is correct. We were perfunctory. All statistical analyses were repeated using non-parametric tests. The details are reported in **Supplementary Tables 1-2**.

It is unclear throughout exactly what statistics were being calculated, but it appeared that in some places there were repeated t-tests performed when the experimental design called for a repeated measures ANOVA (or perhaps the non-parametric version called Friedman's ANOVA) followed by post-hoc testing.

We added the results from a repeated measures ANOVA and report the statistics in the results. We performed a Bonferroni correction and adjusted the statistical significance criteria to $0.05/3 = 0.016$ and followed up individual concentration comparisons using the Wilcoxon rank sum test. These details were clarified in the results and methods.

For the concentration-response functions, the authors should also consider whether they would be better off performing curve fits and comparing the parameters of those fits instead of using concentration by concentration comparisons.

Good idea. We estimated the slope of the input and output concentration-response functions for each preparation and report the Hill coefficient in the results section.

Please report actual values for statistics, including degrees of freedom.

For readability purposes we have included all detailed statistical information in **Supplementary Tables 1-2**.

In the methods it is noted that glomeruli were included in the analysis only if they were present in both the input and the output maps. That seems potentially troubling if part of the input-output transformation is gating the input. Please clarify how many glomeruli were excluded and whether this could influence the analysis.

In some cases the higher sensitivity of the GCaMPs meant that some glomeruli were present in the output activity map, but had no corresponding glomerular peak of activation in the input map. In these cases, including an output glomerulus without a corresponding input glomerulus seemed inappropriate because we didn't know what the corresponding input was doing. This point was clarified in the methods.

Minor Comments

The authors frequently use imprecise terminology in discussing their results that effectively exaggerates their claims. For instance, their data shows increased concentration invariance of neural odor representations but certainly do not provide "direct evidence" that the bulb contributes to the "perception" of concentration invariance.

Good criticism. We have softened the language in several places, and added a sentence in the discussion emphasizing that direct evidence would require a manipulation.

Similarly, in the discussion the authors state that "our results show that odorant identity is more likely determined by the glomerular output of the olfactory bulb," but the present experiments and analyses did not actually compare odor identity representations by comparing different odors.

We have carried out additional experiments from output alone preparations showing that the output maps are odor specific, but relatively concentration invariant. A new figure was added describing this result (**Fig. 5**).

The discussion paragraph about the number of neurons activated seems speculative and inconsistent with the seeming insensitivity of Fura.

It is speculative. But we think it is a useful way to think about our measurements. We have now clearly labeled it as speculative.

The writing of the manuscript needs improvement. The introduction does not adequately consider the existing literature on concentration coding in the olfactory system (consider Stopfer et al. 2003 Identity vs concentration coding in an olfactory system).

References to Stopfer et al., 2003, as well as Niessing and Friedrich 2010 have been added to emphasize that mitral cells were implicated in odor identity coding across a distributed population.

The language of the introduction also is frequently vague (e.g. "a long-standing question in neuroscience")

We have tried to improve the writing of the introduction.

and constantly makes reference to "maps," a term of art among imagers, when it would more effectively refer to spatial patterns of neural activity or patterns of sensory input across glomeruli, etc.

We acknowledge this complaint about the use of maps. However, "maps" is much shorter to write than any of the other suggestions. We have compromised by defining "maps" as spatial patterns of neural activity across glomeruli.

It seems weird to refer to the retina in the abstract and nowhere else. It would be good to discuss the computational utility of concentration invariant representations for downstream regions.

The reference to the retina has been removed, and we added words discussing the utility of such a function to the discussion.

The use of ketamine is tricky for this experiment because it is an NMDAR antagonist and thus could be influencing the input-output relationship. However, it is unclear if NMDARs play a major role in this synapse and alternatives like pentobarbital and isoflurane have their own, possibly more severe problems. I think there should be a caveat in the discussion about this issue.

Good suggestion. We have added a caveat to the discussion.

The scaling of the response maps needs to be made easier for an inexperienced reader to follow, and a grayscale scale bar is needed.

We have added the scalebar and tried to improve the description of the scaling.

Numbering the regions of interest in the maps would help match up the traces.

Numbering for the glomeruli has been added on the ROI panels.

Somewhere specify that the error bars on graphs refer to standard errors.

This was added to the legend of **Fig. 2** and in the methods.

Fig. 2b does not match its caption.

The bar graphs have been replaced with correlation matrices and the wording has been clarified.

If the "antediluvian" microscope has special optics that enable UV imaging of the venerable fura indicator, the methods should indicate that.

No special optics were used. The arc lamp used without any neutral density filters and the described filters, with the camera in its 2x2 binned state with maximal amplifier gain produced dim, but sufficient light that allowed us to image Fura dextran at 340 nm excitation in two preparations. Filter details were added to the methods.

Reviewers' comments:

Reviewer #1 (Remarks to the Author):

In this revision the authors have added some important new experiments using new combinations of indicators to address some of the technical issues raised in the first review and to strengthen their claims about the concentration-dependence of the 'output' and input signals. They have also toned down some of their claims about these differences. The current draft clearly represents a significant effort and their use of such a variety of different reporter combinations is unique and admirable. I remain very skeptical that their claim of higher concentration-invariance among mitral/tufted cells as opposed to sensory inputs is correct, due largely to the fact that the data still rely on epifluorescence imaging and that their experiments still do not appear to cover the dynamic range of the output signal. Nonetheless, I believe that the dataset is comprehensive enough and the analysis is thorough enough for readers to form their own opinion based on what is included here. I do, however, still think that some of the language used is overly strong and that some potential weaknesses in their interpretation are not adequately addressed rhetorically or analytically. Detailed comments below.

1. The conclusion that concentration-dependence in odor representations (or, as the authors put it, "the confound of odor identity and concentration" (p. 12)) is effectively removed by the olfactory bulb is overstated. While the authors' data show that concentration-dependence may be less in the output signal than in the input signal, it is still quite substantial (e.g., glomerular activation patterns appear to change significantly from the lowest to highest concentration in the examples shown in Figs. 4 and 5). They also never formally test the degree to which changing odor concentration interferes with odor identity coding, either at the input or the output side (they only show that pairwise correlations between two odorants are still lower than correlations between different concentrations of the same odorant among output neurons; they do not compare these results with the same analysis for input neurons). They also do not analyze output signal map correlations after attempting to correct for the diffuse signal (which in my opinion is one of the chief sources of their high spatial correlations); instead they perform correlations when measuring only from glomerular regions of interest, but WITHOUT attempting to remove (i.e., by spatial filtering) the diffuse signal. They do analyze concentration-response functions (Fig S5) after such correction, which is important but does not address the issue of correlations in spatial response patterns. In addition, at least one recent imaging study appears to show significant changes in population response patterns of mitral/tufted cells with changes in odor concentration, using 2-photon imaging that excludes lateral dendrite signals. See Economo et al., Neuron 2016, Figure 5A.

Thus in my opinion this general statement should be revised and toned down.

2. Reviewer 4 raised the concern that the analysis included only glomeruli that were present in both the input and output maps, which the authors justified based on signal-to-noise considerations. However this is actually an important analysis as it speaks to the nature of the observed differences in concentration effects between input and output. Indeed, it appears that this may be the chief difference in the input and output maps, based on the

examples shown in Figure 2 and Figs S3 and S4. In other words, as concentration increases, glomeruli 'appear' in the input maps that were already present (and remain present) in the output maps at lower concentrations. The authors could quantify this by counting the fraction of glomeruli that appear in both input and output maps as a function of concentration.

3. A closely related point is the paragraph in the Discussion on p. 15, 'Output signals in the absence of detected input signals'. It is not clear, as written, what the point of the paragraph is other than a technical one about signal detection. But the implications for input-output transformations are important here. First, the authors are implying here (correctly, I believe) that input maps at low concentrations actually include input to glomeruli that appear in the output maps, it is just that these input signals are too weak to be detected (this must be true as there are no known sources of lateral excitation of glomerular output signals). But if they COULD be detected (e.g., with a better activity reporter, etc.), then wouldn't the input maps change less with concentration, and thus show a similar concentration-dependence as do the output maps?

More generally, what is missing from the Discussion is a statement about the inferred nature of the difference in concentration-response functions of input neurons versus mitral/tufted cells that might explain how concentration-invariance in the output arises independent of technical aspects of the measurements. Even if speculative, clarifying this would help the paper appear less phenomenological and hopefully help shape further experiments by the field.

Reviewer #2 (Remarks to the Author):

The authors have addressed my concerns appropriately. The additional experiments and revisions made have further improved the manuscript and strengthened the conclusions.

Perhaps the main question that remains open is whether the difference between input and output signals and maps at different concentrations can somehow be explained by nonlinearities in the optical measurements. Ideally, optical measurements should report neural activity (firing rate) or transmitter release in a linear fashion. In reality, this relationship is probably non-linear, but the exact relationship is unknown. In theory, nonlinearities in optical measurements could create effects that could contribute to the differences between input and output maps. However, for a variety of reasons it appears very unlikely that the observations and conclusions reported in the manuscript are entirely an artifact of such nonlinearities. I therefore believe that the presentation and discussion of the results in this manuscript are appropriate, and that the conclusions are strong.

One way to further address the question raised above is to measure neuronal activity of mitral/tufted cells not in the glomeruli but at the soma using 2-photon microscopy. Because the somatic calcium signal is more likely to reflect action potential output than the dendritic calcium signal, such measurements may reflect output activity more directly. I feel that such measurements would most likely further improve the manuscript, but are not absolutely necessary to merit publication. A potential problem of such measurements is to

assign somatic signals to glomeruli because it may be difficult to follow projections of apical dendrites from somata to glomeruli. Also, it may be very difficult to measure the activity from all output neurons of a glomerulus or a large fraction thereof. If so, direct comparisons between input and output signals of the same glomeruli will be unreliable or impossible. It may also be expected that data from relatively large populations of neurons are required to address the questions raised above, and that mitral and tufted cells behave differently. Addressing all these potential caveats in depth may be a large project. I therefore feel, as said above, that 2-photon measurements of somatic calcium signals would most likely enrich the manuscript but are not absolutely necessary. I do not feel that the impact of the results is diminished substantially by the fact that experiments were performed under anesthesia (although it would of course be interesting to also have results from awake animals).

Reviewer #3 (Remarks to the Author):

The authors added experiments and analyses to address concerns of the other referees, however, the lack of 2-photon imaging of somatic calcium activity still leaves me questioning their "output signal".

Reviewer #4 (Remarks to the Author):

This is a much improved revision of the original manuscript by Storace and Cohen.

1. The main addition is additional experiments comparing the odor-response functions of the sensory neuron input and mitral cell activity in mouse strains expressing the same indicator (GCaMP6f) in each cell population and showing similar results to those previously obtained using the combinations of genetically-encoded calcium indicators and synthetic dyes that they had originally reported. This is a strong solution to the technical concerns raised in the original reviews.

2. It remains unclear exactly how the optical signals from olfactory sensory neurons and mitral cells relate to each other because they are comparing presynaptic calcium flux in the sensory neuron terminals vs some combination of postsynaptic and action potential-evoked dendritic calcium flux in the mitral cells. Another reviewer asked for two-photon imaging of the circuit to help resolve this question, but the authors were unable to provide it because they do not have the necessary equipment. Two-photon imaging would have narrowed the possible interpretations of the wide-field data, but given the new technical analyses ruling out diffuse signals and the like, I don't think it is necessary to support the authors' conclusions.

3. The authors have added a small experiment in which they varied both odor identity and odor concentration to explicitly show that the input-output transformation in this circuit preserves the odor identity representation while shifting towards a more invariant representation of odor concentration. This result is not surprising, but I think it is an important demonstration that helps to contextualize the function of the circuit-level transformation they have discovered.

4. The authors have completely re-done their statistical analyses. They are now correctly

performed and support their claims.

5. The discussion has been improved to more directly address the conclusions and limitations on the study and the inappropriate claims about odor perception have been removed.

6. The authors might consider a title change to more clearly emphasize that the work entails within-animal comparisons.

We are again grateful for the close attention that Reviewer 1 paid to our paper. We think that our response to the reviewer's comments have again improved the paper. There should be some way for us to thank this reviewer as well as the other three reviewers.

Reviewers' comments:

Reviewer #1 (Remarks to the Author):

In this revision the authors have added some important new experiments using new combinations of indicators to address some of the technical issues raised in the first review and to strengthen their claims about the concentration-dependence of the 'output' and input signals. They have also toned down some of their claims about these differences. The current draft clearly represents a significant effort and their use of such a variety of different reporter combinations is unique and admirable. I remain very skeptical that their claim of higher concentration-invariance among mitral/tufted cells as opposed to sensory inputs is correct, due largely to the fact that the data still rely on epifluorescence imaging and that their experiments still do not appear to cover the dynamic range of the output signal. Nonetheless, I believe that the dataset is comprehensive enough and the analysis is thorough enough for readers to form their own opinion based on what is included here. I do, however, still think that some of the language used is overly strong and that some potential weaknesses in their interpretation are not adequately addressed rhetorically or analytically. Detailed comments below.

1. The conclusion that concentration-dependence in odor representations (or, as the authors put it, "the confound of odor identity and concentration" (p. 12)) is effectively removed by the olfactory bulb is overstated. While the authors' data show that concentration-dependence may be less in the output signal than in the input signal, it is still quite substantial (e.g., glomerular activation patterns appear to change significantly from the lowest to highest concentration in the examples shown in Figs. 4 and 5).

We agree. We have added words in the discussion saying that the output maps, while more similar than the input maps, are not identical. Perhaps other brain regions would remove even these differences. Or pattern completion elsewhere might mean that this level of identity is good enough.

They also never formally test the degree to which changing odor concentration interferes with odor identity coding, either at the input or the output side (they only show that pairwise correlations between two odorants are still lower than correlations between different concentrations of the same odorant among output neurons; they do not compare these results with the same analysis for input neurons).

We added a paragraph to the discussion indicating that although our analysis indicates that the output response is more highly correlated with the same odor at different concentrations than two different odorants, this result is likely to depend on the chosen odorants. In our example we selected odorants with very distinct patterns to highlight the effect. However, it's plausible that two different odorants with similar glomerular activation patterns might be better correlated at some concentrations than the same odorant at different concentrations.

They also do not analyze output signal map correlations after attempting to correct for the diffuse signal (which in my opinion is one of the chief sources of their high spatial correlations); instead they perform correlations when measuring only from glomerular regions of interest, but WITHOUT attempting to remove (i.e., by spatial filtering) the diffuse signal.

Good idea. We performed this analysis on the output maps. High-pass spatial filtering tended to slightly (but non-significantly) reduce the output correlation coefficients. This result is included in the Results section.

They do analyze concentration-response functions (Fig S5) after such correction, which is important but does not address the issue of correlations in spatial response patterns. In addition, at least one recent imaging study appears to show significant changes in population response patterns of mitral/tufted cells with changes in odor concentration, using 2-photon imaging that excludes lateral dendrite signals. See Economo et al., Neuron 2016, Figure 5A. Thus in my opinion this general statement should be revised and toned down.

The data presentation in Economo et al. have been adjusted to the same scale rather than each map using its own scale. This makes it difficult to compare maps at different concentrations. However, we have added words to the discussion emphasizing that the output maps do change, but just less than the input maps.

2. Reviewer 4 raised the concern that the analysis included only glomeruli that were present in both the input and output maps, which the authors justified based on signal-to-noise considerations. However this is actually an important analysis as it speaks to the nature of the observed differences in concentration effects between input and output. Indeed, it appears that this may be the chief difference in the input and output maps, based on the examples shown in Figure 2 and Figs S3 and S4. In other words, as concentration increases, glomeruli 'appear' in the input maps that were already present (and remain present) in the output maps at lower concentrations. The authors

could quantify this by counting the fraction of glomeruli that appear in both input and output maps as a function of concentration.

We agree this type of analysis is a good idea although as previously noted we were concerned about signal-to-noise concerns and would prefer to wait until these measurements are performed using 2-photon imaging.

3. A closely related point is the paragraph in the Discussion on p. 15, 'Output signals in the absence of detected input signals'. It is not clear, as written, what the point of the paragraph is other than a technical one about signal detection. But the implications for input-output transformations are important here. First, the authors are implying here (correctly, I believe) that input maps at low concentrations actually include input to glomeruli that appear in the output maps, it is just that these input signals are too weak to be detected (this must be true as there are no known sources of lateral excitation of glomerular output signals). But if they COULD be detected (e.g., with a better activity reporter, etc.), then wouldn't the input maps change less with concentration, and thus show a similar concentration-dependence as do the output maps?

We don't agree. The difference in concentration sensitivity cannot be explained by signal-to-noise differences. The main results are that the activity of the output glomeruli are more similar across concentrations, and that the input is more concentration-dependent than the output.

For example, in **Figure S3**. Input signals were only detected from a single glomerulus at 0.36% saturated vapor. Even with an ideal sensor, the signals from the other glomeruli would be much smaller than the signal from the detected glomerulus. Furthermore, at 0.12% there are no detected signals. If we could detect them with an ideal sensor, the signal would still be much smaller than the signals at a higher concentration. Thus, even with an ideal sensor the input maps would be less similar to each other across concentrations, and more concentration dependent than the output.

More generally, what is missing from the Discussion is a statement about the inferred nature of the difference in concentration-response functions of input neurons versus mitral/tufted cells that might explain how concentration-invariance in the output arises independent of technical aspects of the measurements. Even if speculative, clarifying this would help the paper appear less phenomenological and hopefully help shape further experiments by the field.

We have cited several papers that have demonstrated plausible mechanisms that could contribute to generating stable output maps, including various interneurons that are involved in gain control (e.g., Zhu et al., 2013, Kato et al., 2013; Miyamichi et al., 2013; Banerjee et al., 2015). We have also cited

Cleland who has proposed that the bulb likely performs a relational normalization via specific bulb interneuron cell types. Beyond this, one of the authors (LBC) is not comfortable about speculating about the mechanism because of the large number of interneuron types in the bulb, and the inputs from many higher brain centers which may have a role in the transformation we have measured.

Reviewer #2 (Remarks to the Author):

The authors have addressed my concerns appropriately. The additional experiments and revisions made have further improved the manuscript and strengthened the conclusions.

Perhaps the main question that remains open is whether the difference between input and output signals and maps at different concentrations can somehow be explained by nonlinearities in the optical measurements. Ideally, optical measurements should report neural activity (firing rate) or transmitter release in a linear fashion. In reality, this relationship is probably non-linear, but the exact relationship is unknown. In theory, nonlinearities in optical measurements could create effects that could contribute to the differences between input and output maps. However, for a variety of reasons it appears very unlikely that the observations and conclusions reported in the manuscript are entirely an artifact of such nonlinearities. I therefore believe that the presentation and discussion of the results in this manuscript are appropriate, and that the conclusions are strong.

One way to further address the question raised above is to measure neuronal activity of mitral/tufted cells not in the glomeruli but at the soma using 2-photon microscopy. Because the somatic calcium signal is more likely to reflect action potential output than the dendritic calcium signal, such measurements may reflect output activity more directly. I feel that such measurements would most likely further improve the manuscript, but are not absolutely necessary to merit publication. A potential problem of such measurements is to assign somatic signals to glomeruli because it may be difficult to follow projections of apical dendrites from somata to glomeruli. Also, it may be very difficult to measure the activity from all output neurons of a glomerulus or a large fraction thereof. If so, direct comparisons between input and output signals of the same glomeruli will be unreliable or impossible. It may also be expected that data from relatively large populations of neurons are required to address the questions raised above, and that mitral and tufted cells behave differently. Addressing all these potential caveats in depth may be a large project.

We agree. We have suggested doing such experiments in grant proposals and consider them an important next step. However, because of the caveats mentioned by the reviewer we expect this to be a large effort.

I therefore feel, as said above, that 2-photon measurements of somatic calcium signals would most likely enrich the manuscript but are not absolutely necessary. I do not feel that the impact of the results is diminished substantially by the fact that experiments were performed under anesthesia (although it would of course be interesting to also have results from awake animals).

Reviewer #3 (Remarks to the Author):

The authors added experiments and analyses to address concerns of the other referees, however, the lack of 2-photon imaging of somatic calcium activity still leaves me questioning their "output signal".

Reviewer #4 (Remarks to the Author):

This is a much improved revision of the original manuscript by Storace and Cohen.

1. The main addition is additional experiments comparing the odor-response functions of the sensory neuron input and mitral cell activity in mouse strains expressing the same indicator (GCaMP6f) in each cell population and showing similar results to those previously obtained using the combinations of genetically-encoded calcium indicators and synthetic dyes that they had originally reported. This is a strong solution to the technical concerns raised in the original reviews.

2. It remains unclear exactly how the optical signals from olfactory sensory neurons and mitral cells relate to each other because they are comparing presynaptic calcium flux in the sensory neuron terminals vs some combination of postsynaptic and action potential-evoked dendritic calcium flux in the mitral cells.

We agree. A clearer experiment would be using a voltage sensor in the input and output cells where the range of voltage sensitivity only included voltage signals present in action potentials. However, the sensor needs to be targeted to the nerve terminal in the input so that it doesn't measure action potential signals in the axons approaching the glomerulus only the action potentials in the glomerulus. And it would be better if the output sensor targeted the dendritic tuft in the glomerulus. Such targeted sensors do not exist although we hope to develop them in the future.

Another reviewer asked for two-photon imaging of the circuit to help resolve this question, but the authors were unable to provide it because they do not have the necessary equipment. Two-photon imaging would have narrowed the possible

interpretations of the wide-field data, but given the new technical analyses ruling out diffuse signals and the like, I don't think it is necessary to support the authors' conclusions.

3. The authors have added a small experiment in which they varied both odor identity and odor concentration to explicitly show that the input-output transformation in this circuit preserves the odor identity representation while shifting towards a more invariant representation of odor concentration. This result is not surprising, but I think it is an important demonstration that helps to contextualize the function of the circuit-level transformation they have discovered.

4. The authors have completely re-done their statistical analyses. They are now correctly performed and support their claims.

5. The discussion has been improved to more directly address the conclusions and limitations on the study and the inappropriate claims about odor perception have been removed.

6. The authors might consider a title change to more clearly emphasize that the work entails within-animal comparisons.

We tried to think of a better title but did not succeed.

REVIEWERS' COMMENTS:

Reviewer #1 (Remarks to the Author):

The authors have responded to my earlier comments and the remaining concerns of the other reviewers; mainly with additional qualifications in the text, although with a few new analyses. There seems to be consensus among the reviewers about the lingering weaknesses in the study, but also consensus that the authors have adequately addressed the major concerns. Thus, I have no additional issues to raise.

Reviewer #4 (Remarks to the Author):

In this revised manuscript, the authors have adequately addressed all of the previous concerns raised by the reviewers. The technical concerns have been addressed as far as they can be within the scope of the project (i.e. with new analyses rather than repeating the project with 2-photon imaging), and the scientific conclusions are well-founded. The limitations on data interpretation raised by the reviewers have been appropriately incorporated into the language of the paper and its discussion section. This paper is a substantial contribution to the literature on olfactory information processing.

Response to reviewers.

Neither of the reviewers raised additional issues. We again want to express our gratitude for the consistently constructive and thoughtful comments and suggestions made by all four reviewers, which dramatically improved our manuscript.

Reviewer #1 (Remarks to the Author):

The authors have responded to my earlier comments and the remaining concerns of the other reviewers; mainly with additional qualifications in the text, although with a few new analyses. There seems to be consensus among the reviewers about the lingering weaknesses in the study, but also consensus that the authors have adequately addressed the major concerns. Thus, I have no additional issues to raise.

Reviewer #4 (Remarks to the Author):

In this revised manuscript, the authors have adequately addressed all of the previous concerns raised by the reviewers. The technical concerns have been addressed as far as they can be within the scope of the project (i.e. with new analyses rather than repeating the project with 2-photon imaging), and the scientific conclusions are well-founded. The limitations on data interpretation raised by the reviewers have been appropriately incorporated into the language of the paper and its discussion section. This paper is a substantial contribution to the literature on olfactory information processing.

We agree!